# Understanding the interplay of GDP, renewable, and non-renewable energy on carbon emissions: Global wavelet coherence and Granger causality analysis

**Yuganthi Caldera[1], Tharulee Ranthilake[1], Heshan Gunawardana[1], Dilshani Senevirathna[1], Ruwan Jayathilaka[2]\*, Nilmini Rathnayake[1], Suren Peter[1]**

**1** SLIIT Business School, Sri Lanka Institute of Information Technology, Malabe, Sri Lanka, **2** Department of Information Management, SLIIT Business School, Sri Lanka Institute of Information Technology, Malabe, Sri Lanka

\* ruwan.j@sliit.lk

**Data Availability Statement:** The data underlying the results presented in the study are available from World Bank Indicators (https://data.worldbank.org/indicator) data sets. 1) CO2

## Abstract

This study examines the causality of Per Capita Gross Domestic Production (PGDP), Renewable Energy Consumption (REC), and Non-Renewable Energy Consumption (NREC) on Carbon dioxide ($CO_2$) emissions at the global level utilising data gathered from 1995 to 2020 across various countries categorised based on income levels as High, Low, Upper Middle and Lower Middle and analysed through wavelet coherence. The findings reveal both bidirectional and unidirectional causality between the variables which have evolved. Globally, a bi-directional relationship is observed with a positive correlation between PGDP and NREC and in contrast, a negative correlation with REC. Furthermore, the analysis highlights varying causalities between $CO_2$ emissions and PGDP, except for high-income and lower-middle-income country categories, all other shows one-way causality in different periods in the short term. Moreover, $CO_2$ and REC, show unidirectional causality throughout the short-term, exceptionally medium & long term have both unidirectional and bidirectional causalities across all country categories with a positive correlation. In contrast, $CO_2$ and NREC depict similar causalities to REC, however, with a negative correlation. A cross-country analysis was performed between $CO_2$ and PGDP, $CO_2$ and REC, and $CO_2$ and NREC using Granger causality which shows mixed relationships. The findings hold significant implications for policymakers, providing valuable insights into the trade-offs between economic growth, energy consumption, and carbon emissions.

## Introduction

Currently, the world faces a pressing challenge in the form of extreme climate fluctuations driven by global warming, with $CO_2$ emerging as a prominent contributor among various greenhouse gases [1–6]. $CO_2$, primarily emitted from activities such as fossil fuel combustion, cement manufacturing and transportation stands out as the foremost contributor to global

emission in Metric tons Per Capita - https://data.worldbank.org/indicator/EN.ATM.CO2E.PC 2) Per capita GDP (Current US$) - https://data.worldbank.org/indicator/NY.GDP.PCAP.CD 3) % of total final energy consumption - https://data.worldbank.org/indicator/EG.FEC.RNEW.ZS 4) 100 - % of total final energy consumption - https://data.worldbank.org/indicator/EG.FEC.RNEW.ZS.

**Funding:** The author(s) received no specific funding for this work.

**Competing interests:** The authors have declared that no competing interests exist.

warming. Its potent heat-trapping properties and significant concentration within the Earth's atmosphere contribute to the escalating impacts of climate change [7]. Understanding the sources and dynamics of $CO_2$ emission is crucial for devising effective strategies to mitigate climate change and its adverse effects.

As economies expand, the demand for energy rises, often leading to increased reliance on fossil fuels for energy production. This growth-driven rise in energy consumption is closely linked to rising $CO_2$ emissions, posing significant challenges to achieving sustainable development [8]. Particularly in developing nations, non-renewable energy sources are often overused or mismanaged due to various socio-economic factors. Consequently, addressing the dual imperative of economic growth and environmental sustainability has become a focal point for both researchers and policymakers worldwide. Studies exploring the Environmental Kuznets Curve hypothesis have provided valuable insights into the complex relationship between economic growth and environmental degradation. The same relationship is confirmed by studies conducted for 11 newly developed countries and 83 selected countries as concrete evidence for $CO_2$ emission and GDP growth [9, 10]. However, regulating $CO_2$ emissions presents a multifaceted challenge due to its close association with energy production. As a result, preventive measures targeting $CO_2$ emissions could potentially hinder economic growth, particularly in developing nations [11]. Numerous empirical studies across diverse nations have consistently revealed a direct correlation between economic growth and $CO_2$ emissions by utilizing sophisticated econometric techniques for countries such as Pakistan [12], China [13], Kazakhstan [14], Indonesia [15], Turkey [16], Nigeria [17], Egypt [11], and Bangladesh [18].

Renewable energy sources, including wind, solar, geothermal, wave and tidal power are anticipated to be the fastest-growing sectors within the energy industry. This growth is driven by the recognition of renewable energy's crucial role in addressing energy security and climate change issues. In Malaysia, several studies have implied the significant role of natural gas consumption in driving economic growth while promoting environmental sustainability. Natural gas, comparing to oil and coal, emits lower levels of $CO_2$, positioning it as a crucial energy source for mitigating environmental degradation and supporting transitions towards cleaner energy [19]. Notably, China's substantial investment of over 170 billion USD in clean energy surpasses the European Union and the United States, signalling a significant shift in global energy investments [20]. This investment highlights the pivotal role of renewable energy in driving economic development and reducing greenhouse gas emissions on a global scale. Despite the promising potential of renewable energy consumption, its widespread adoption is expected to bring both opportunities and challenges. While renewable energy consumption holds promise for meeting future energy demands and reducing carbon dioxide emissions, it also poses challenges such as the need for infrastructure development and grid integration. Additionally, emerging technologies like hydrogen or electronic (battery-driven) based vehicles present promising alternatives to traditional fossil fuels powered transportation, however, their widespread adoption requires overcoming technical and economic barriers [21–23].

The main objective of this study is to examine the causality of the variables $CO_2$, PGDP, REC and NREC considering variations over a 26-year period in order to address a significant knowledge gap in the existing literature. Understanding the causality enhances the novelty of our study by revealing the intricate relationship between economic growth, renewable and non—renewable energy consumption, and $CO_2$ emissions trough sophisticated techniques like wavelet coherence and Granger causality analysis. This approach transcends simple correlation, offering a dynamic perspective on how these variables interact over time. By differentiating between unidirectional and bidirectional causality patterns and examining dynamics across global income levels, our study fills a substantial gap in the literature. It equips policy makers with actionable insights to design targeted interventions for sustainable development

and climate mitigation efforts. This comprehensive understanding of causality dynamics significantly enriches the existing body of knowledge and enhances the effectiveness of policy making in addressing pressing environmental challenges. Further, researchers intend to conduct an income category wise analysis to investigate variations between different country categories, representing a recent addition to the literature. Thus, this study shows a prominent value addition in three keyways.

Firstly, researchers employ a novel methodology of Wavelet coherence, which provides a unique perspective on the relationship and causality between $CO_2$, PGDP, REC and NREC. Unlike previous studies, Wavelet coherence allows a more nuanced analysis by simultaneously considering both time and frequency domains, providing insights into the dynamic interactions between these variables over time. This methodology helps to examine the linkage between the variables across short-term, medium-term and long-term time scales with the direction of arrows for the period of 1995 to 2020. Further, the methodology shows correlation variations emphasising whether it is high, medium or low between variables with the frequency scale as well. Additionally, Granger causality was adopted to analyse the causality between the variables for each individual country to enhance the findings shown through wavelet coherence.

Secondly, this study adopts a global approach covering 161 countries categorised based on the income level. While existing literature has predominantly focused on individual countries, organisations, and different regions, this comprehensive global study based on income level provides a unique insight into the relationship between economic growth, energy consumption, and $CO_2$ emissions.

Thirdly, the study presents a summarisation of graphs plotted for the world and all income categories through wavelet coherence into one output. This output showcases the directions and relationships between variables across different time scales; short term, medium term and long term, highlighting correlation variations over the 26-year period by allocating a five-year time range. Hence, this study offers valuable insights for policymakers, researchers, and stakeholders in understanding and addressing the complex dynamics of $CO_2$ emissions and energy consumption.

Further, it sheds light on the role of technological advancements in reducing $CO_2$ emissions and the impact of increased utilisation of renewable energy sources on mitigating $CO_2$ emissions. This research serves as a resource for inventors, policymakers, and entrepreneurs, providing them with predictive insights into how innovations in technology and policy can contribute to the reduction of $CO_2$ emissions and the preservation of the environment. Through a comprehensive understanding of the factors driving $CO_2$ emissions and the potential solutions available, stakeholders can make informed decisions and take proactive steps toward promoting sustainability and mitigating climate change.

The subsequent sections of the article are structured as follows: a review of existing literature, followed by an explanation of the data and methodology, then the presentation of findings and discussion, and finally, the overall conclusion of the study.

## Literature review

Fossil fuel combustion is a major contributor to the global issue of climate change, releasing a large amount of $CO_2$ emissions [24]. A similar study done using panel Granger causality analysis across various income categories such as high-income, upper-middle-income, lower-middle-income, and low-income has identified bidirectional causality between $CO_2$ emission and energy consumption. It further reveals a unidirectional causality between GDP and $CO_2$ emission in upper-middle-income economies, A unidirectional causality is identified in lower-

middle-income economies and both the long and short run, a bidirectional causality is displayed in low-income economies [25, 26]. Furthermore, the connection between non-renewable energy and $CO_2$ emission is shown in the results that changes in $CO_2$ emission will lead to changes in non-renewable energy consumption with a negative relationship [27]. Additionally, analyses conducted on G-20 countries between 2010 and 2019 have highlighted the positive and significant impact of GDP and non-renewable energy consumption on $CO_2$ emissions, whereas a negative and significant effect was observed from renewable energy sources [28]. Notably, despite the growing body of literature in this area, only a few studies have examined income level based categorisation of economies using wavelet methodology.

A study conducted in Saudi Arabia investigated the consumption of non–renewable energy and its associated factors, which have contributed to an increase in $CO_2$ emissions [29]. Moreover, the study examines the relationship between the income level of a country and its $CO_2$ emission, and it confirms the Environmental Kuznets Curve hypothesis. This hypothesis suggests that increased income can initially lead to higher carbon emissions, and it is possible to mitigate these emissions without compromising economic growth [30]. Additionally, trade openness helps to reduce pollution levels and highlights the need for advanced technologies and renewable energy sources in environmental management. A study utilising wavelet coherence to investigate the relationship between economic growth and environmental pollution found that G7 countries exhibit higher coherence in the short term, with increased economic growth correlating with higher pollution levels [31]. The study suggests that G7 countries should strengthen economic cooperation and design efficient policy instruments, such as implementing short-term taxation on those who emit carbon dioxide on a mass scale in order to address the environmental challenges.

In identifying the asymmetric causality in greenhouse gas emissions in Saudi Arabia, it was found that, unidirectional asymmetric causality results from both positive and negative changes in $CO_2$ emissions to REC [32]. In both the short-term and long-term periods, the same outcome of asymmetric causality resulting from the positive and negative shocks of the real GDP to REC occurred. Similarly, analyses conducted in Algeria, a country which is rich in renewable energy resources, highlight the low share of renewable energy utilization despite its abundance. Studies in this context demonstrate a significant correlation between economic growth and $CO_2$ emission [33]. Furthermore, wavelet coherence results indicate that economic growth is the leading variable influencing both $CO_2$ emission and energy consumption suggesting a co-movement between economic growth and $CO_2$ emissions. Interestingly, economic growth appears to have a negative impact on energy consumption but positively affects $CO_2$ emissions, as observed in Sub-Saharan Africa [34]. These studies suggest that $CO_2$ emissions may stimulate economic growth and they do not necessarily drive energy consumption, highlighting the complex dynamics between economic development, energy use, and environmental outcomes.

Countries like Oman, Qatar, and Saudi Arabia show an inverted U-shaped Environmental Kuznets Curve, where $CO_2$ emissions initially increase with GDP growth before eventually declining [35]. To address this trend, it is recommended to implement necessary economic and social policies that are aimed at reducing $CO_2$ emissions while simultaneously fostering economic growth. Similarly, a study conducted in Thailand using Wavelet and Granger analysis techniques showed that economic growth leads to $CO_2$ emission, and energy consumption and $CO_2$ emissions mutually predict each other [36]. The same findings have strengthened the awareness of energy efficiency among citizens to minimise the negative impacts of $CO_2$ emissions on the environment. In Ghana, the energy sector heavily relies on fossil fuels which have a detrimental effect on $CO_2$ emissions [37]. The results of the causality studies support a unidirectional relationship between energy consumption and economic growth. The level of

penetration of renewable energy has not reached the point to mitigate $CO_2$ emissions. Consequently, Ghanaian policymakers should implement policies focused on sustainable development, such as increasing the usage of renewable energy.

The reliance on coal, oil, and natural gas has emerged as the primary driver of non-renewable energy consumption, posing sustainability challenges due to the finite nature of these resources [38]. Continuous reliance on non-renewable energy sources risks resource depletion in the long term. Moreover, empirical findings suggest a bidirectional causality between renewable and non-renewable energy sources and economic growth in the short and long run [39]. Therefore, managing the use of non-renewable energy sources may not be sufficient to foster sustainable economic growth.

In culmination of the literature review, critical lacunae emerge in the existing body of research, notably pertaining to the dearth of studies that systematically categorise economic pathways across income levels using wavelet methodologies. This gap underscores the necessity for comprehensive analyses that not only employ advanced analytical techniques but also encompass diverse country contexts to elucidate the exact dynamics between economic growth, energy consumption and $CO_2$ emissions. Furthermore, despite the considerable scholarly attention devoted to the Environmental Kuznets Curve hypothesis and the relationship between economic growth and $CO_2$ emissions, there persists a notable absence of comprehensive assessments that encompass temporal dynamics and diverse country contexts. Additionally, the literature review reveals a conspicuous gap in the examination of the role of renewable energy utilization in mitigating $CO_2$ emissions, particularly in regions abundant in renewable resources. Addressing these gaps is imperative to inform evidence-based policy interventions aimed at fostering sustainable development and mitigating adverse effects of climate change.

## Data and methodology

### Data

The study encompasses annual data collected from 161 countries spanning the years 1995 to 2020. Data have been extracted through secondary data presented in S1 Appendix, outlining the variables of interest. The relevant source and measure of the variables taken are illustrated in Table 1.

The selected 161 countries comprise of 45 high-income countries, 20 low-income Countries, 51 low-middle-income countries, and 45 upper-middle-income countries to have a comprehensive analysis to investigate the causality among the variables $CO_2$, PGDP, REC and NREC.

### Methodology

The study employs Wavelet coherence methodology to assess causality and correlation within the time series dataset utilising R-Software. The inception of this methodology is explained by

**Table 1. Data sources and definition of variables.**

| Variables | Measure | Source |
|---|---|---|
| $CO_2$ Emissions | Metric tons Per Capita of $CO_2$ | https://data.worldbank.org/indicator/EN.ATM.CO2E.PC |
| PGDP | Per capita GDP (Current US$) | https://data.worldbank.org/indicator/NY.GDP.PCAP.CD |
| REC | % of total final energy consumption | https://data.worldbank.org/indicator/EG.FEC.RNEW.ZS |
| NREC | 100 - % of total final energy consumption | https://data.worldbank.org/indicator/EG.FEC.RNEW.ZS |

Source: Authors' Compilation.

P. Goupillaud [40], and currently, the wavelet approach has been discussed in many studies [33, 36]. Fourier analysis is the root of the Wavelet coherence method, which is used to obtain results through graphical representations, but it is not an efficient way to detect sudden fluctuations. Nevertheless, the Wavelet approach has overcome these limitations with the existence of a zero mean based on a limited period [41, 42].

This section serves to introduce the foundational aspects and expansion of the current study model. The foundation of current study model is grounded in previous research findings, guiding our expectations regarding the signs of relationships between independent variables and dependent variables. Specifically, Table 2 comprises the past studies emphasising the expected sign of the variables.

A Wavelet, generated by the function $\psi^{a,b}(x)$, with contractions, $a$ and translation, $b$ can be statistically denoted as follows ($\psi$ denotes Morlet Wavelet function),

$$\psi^{a,b}(x) = |a|^{-\frac{1}{2}} \psi \left( \frac{x-b}{a} \right) \tag{1}$$

Previous studies have emphasised the advantages of using wavelet coherence methodology. Data filling is not required unlike other linear and nonlinear methodologies, and results can be obtained through graphical representation without relying on numerical statistics. Moreover, the fluctuations of the peri can also be categorised based on long term, medium-term, and short-term and based on the frequencies as well [53]. Considering finance and economics, medicine, tourism and financial development, the Wavelet coherence approach has made a prominent step [54, 55]. The Wavelet coherence method is a bivariate analysis which explains why the other variables lead the second with different time ranges. This is more advanced than approaches such as Granger causality since it can explain the direction and the strength of the causal relationship between the variables. Two-time series can be evaluated in wavelet to realise what variable impacts the other.

As mentioned above in previous empirical studies, wavelet analysis is significant in time-frequency analysis which means it provides the frequency in time representations for the causality of variables. Moreover, this provides multiscale analysis which can capture the causality of variables with different frequencies.

The Granger causality test was adopted for each individual country to enhance the results gained through Wavelet coherence analysis [56]. This study uses VAR Granger analysis to evaluate the causality between $CO_2$ and PGDP, $CO_2$ and REC, and $CO_2$ and NREC for each country utilizing Stata software. This approach has been done in many studies [57]. First, the study ran the Augmented Dickey-Fuller test to check the stationarity of the variables of the two-time series. The two stationary covariance variables are X and Y. This shows the causality for each individual, explaining that the Xi′t variable causes Yi, t if it's better able to predict Yi,t

**Table 2. Variables and supporting past studies.**

| Variable | Relationship | Past Studies |
|---|---|---|
| $CO_2$ and PGDP | Bidirectional— Positive | Attiaoui, Toumi [43]; Peng, Tan [44]; Mohamed Yusoff, Ridzuan [19]; Dharmapriya, Edirisinghe [45] Raihan, Voumik [46]; Banday and Aneja [47] |
| $CO_2$ and REC | Bidirectional— Negative | Le [27]; Attanayake, Wickramage [48]; Ponce and Khan [49] |
| $CO_2$ and NREC | Bidirectional— Positive | Phatchapa Boontome [50]; Dogan and Seker [51]; Dogan and Seker [52] |

Source: Authors' Compilation.

by employing all the required information of variables, which is compared to the use of information except Xi,t for each individual. The assumption would be the model is linear, therefore using the time-stationary VAR representation to each cross-sectional unit i and time period t.

$$Y_{i,t} = \sum_{k=1}^{p} \beta_i \, Y_{i,t-k} + \sum_{k=0}^{p} \theta_k \, X_{i,t-k} + u_{i,t} \tag{2}$$

There $u$ is normally distributed with $u_{i,t} = \alpha_i + \varepsilon_{i,t}$, where the number of lags is denoted as $p$. Under the assumption the autoregressive coefficients $\beta_k$ and the regression coefficients $\theta_k$'s are constant for $k \, \epsilon \, [1, N]$. The equation depicts that u is normally distributed, where Y is the dependent variable (i and t denote the country and time, respectively), X is the independent variable, ui, t denotes the error term, and k is the number of lags.

## Results and discussion

The descriptive statistics of the current study are presented in Table 3. The dataset comprises 4186 observations, with 1170, 520, 1170, and 1326 observations corresponding to high-income, low-income, upper-middle-income, and lower-middle-income groups respectively. The average $CO_2$ emissions at the global level for, high income, low-income, upper-middle-

**Table 3. Summary of descriptive statistics of variables.**

| Country Category | | Variables | | | |
|---|---|---|---|---|---|
| | | CO$_2$ | PGDP | REC | NREC |
| **Global** | Obs | 4186 | 4186 | 4186 | 4186 |
| | Mean | 3.967 | 11358.670 | 34.877 | 65.123 |
| | SD | 5.175 | 19489.530 | 30.428 | 30.428 |
| | Min | 0.022 | 99.757 | 0.002 | 1.660 |
| | Max | 47.657 | 179467.500 | 98.340 | 99.998 |
| **High Income** | Obs | 1170 | 1170 | 1170 | 1170 |
| | Mean | 9.181 | 33188.830 | 16.503 | 83.497 |
| | SD. | 6.636 | 26009.210 | 17.060 | 17.060 |
| | Min | 1.355 | 826.973 | 0.010 | 17.210 |
| | Max | 47.657 | 179467.500 | 82.790 | 99.990 |
| **Lower Income** | Obs | 520 | 520 | 520 | 520 |
| | Mean | 0.266 | 709.851 | 75.601 | 24.399 |
| | SD | 0.525 | 1169.685 | 26.689 | 26.689 |
| | Min | 0.022 | 99.757 | 0.580 | 1.660 |
| | Max | 3.099 | 11304.640 | 98.340 | 99.420 |
| **Upper Middle Income** | Obs | 1170 | 1170 | 1170 | 1170 |
| | Mean | 3.445 | 5176.533 | 21.475 | 78.525 |
| | SD. | 2.730 | 3137.035 | 19.281 | 19.281 |
| | Min | 0.470 | 252.975 | 0.002 | 9.880 |
| | Max | 15.341 | 19849.720 | 90.120 | 99.998 |
| **Lower Middle Income** | Obs | 1326 | 1326 | 1326 | 1326 |
| | Mean | 1.279 | 1727.602 | 46.945 | 53.055 |
| | SD | 1.485 | 1337.658 | 28.413 | 28.413 |
| | Min | 0.050 | 107.393 | 0.060 | 5.230 |
| | Max | 7.751 | 9225.845 | 94.770 | 99.940 |

Note: Obs, SD, MIN, and MAX define Observation, Standard Deviation, Minimum, and Maximum, respectively

Source: Authors' Compilation.

income, and lower-middle-income groups are 3.97 metric tons per capita, 9.181 metric tons per capita which is the highest, 0.266 metric tons per capita the lowest, 3.445 metric tons per capita, and 1.279 metric tons per capita respectively. The average values of PGDP are US\$ 11,359, US\$ 33,189 the highest, US\$ 710 the lowest, US\$ 5,177, and US\$ 1,728 at the global level for, high-income, low-income, upper-middle-income, and lower-middle-income respectively. Considering REC and NREC, it would calculate the 100% of final energy consumption, and the average values of REC are 34.87% for the world, 16.5% for high income, 75.6% for low income, 21.48% for upper-middle income, and 46.94% for lower-middle income. This suggests that low middle-income countries have the highest proportion of renewable energy consumption, while high-income countries have the lowest.

The violin plots in Fig 1 depict the dispersion of the data for $CO_2$ emission, PGDP, RE, and NRE for all four country categories, namely high income, low income, lower middle-income, and upper middle-income. The data are clustered together for the $CO_2$ emission of all the country categories except for lower-middle-income countries, indicating less variation among the data is less. The data points are distributed around the mean for low-income countries.

Considering the data dispersion of data for RE, similar to $CO_2$ the data has been clustered together for high income countries and upper middle-income countries. The distribution of data points around the mean is observed for low-income countries, while data is distributed below the mean for lower-middle-income countries. In RE, the median of the low-income county category is higher than other country categories, and vice versa for NRE. Further, in lower-middle-income countries, there is a uniform distribution, while in high-income and upper-middle-income countries, the distribution is below the mean. The NRE shows the exact opposite results to RE due to the perfect correlation between the two variables.

The wavelet coherence graphs for the world and four income categories have been discussed with interpretations in this section. Table 4 depicts the attributes related to the graph and how they have been interpreted. In this study, the first variable is considered $CO_2$, and the second variable is PGDP, REC, and NREC. The graphs have been plotted as $CO_2$ and PGDP, $CO_2$ and REC, and $CO_2$ and NREC for the global level and income category-wise.

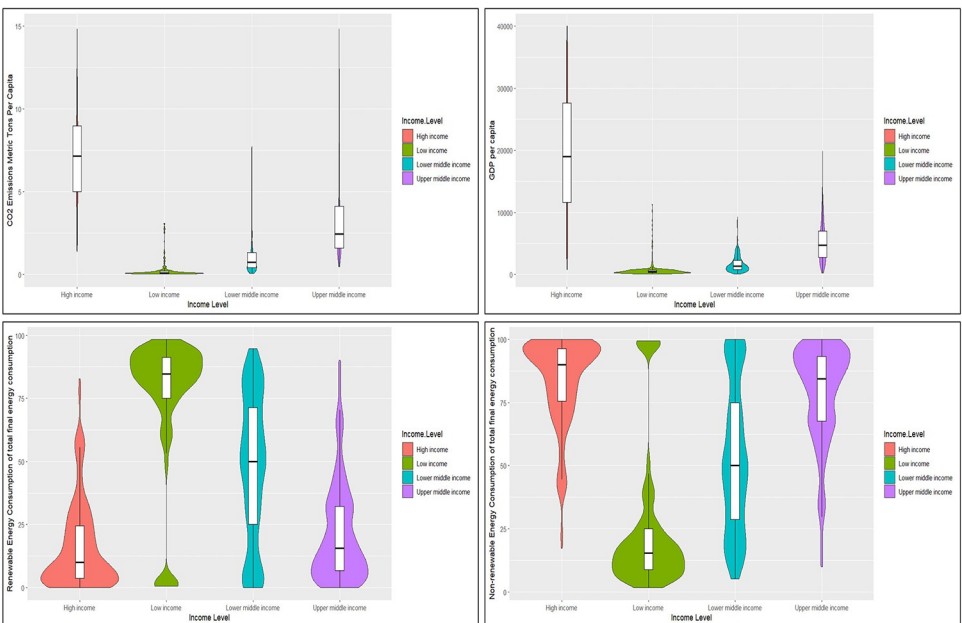

**Fig 1. Violin plots of CO2 emission, GDP, RE and NRE by country category.** Source: Authors' Compilation.

**Table 4. Interpretation of wavelet coherence.**

| Attributes | Interpretation |
| --- | --- |
| Horizontal Axis | Time period |
| Vertical Axis | Scale |
| White Cone | Cone of Influence |
| Thick Black border | 95% confidence level |
| Rightward arrow | Positive relationship (In-Phase) |
| Leftward arrow | Negative relationship (Out-phase) |
| Rightward up arrow | Second variable (PGDP/RE/NREC) causes the first variable ($CO_2$) |
| Rightward down arrow | First Variable ($CO_2$) causes the second Variable (PGDP/RE/NREC) |
| Leftward up arrow | Second variable (PGDP/RE/NREC) causes the first variable ($CO_2$) |
| Leftward down arrow | First Variable ($CO_2$) causes the second Variable (PGDP/RE/NREC) |
| Cold region (Blue) | No correlation |
| Warm region (Red) | Correlation exists |

Source: Authors' Compilation.

The white cone of the illustrations depicts the Cone of Influence which means the region in the time-frequency where the edge effects are significant. The 5% significance level is interpreted using the thick black border calculated using the Monte Carlo simulations. The red region signifies that there is a correlation, and the blue region explains that the correlation doesn't exist. Moreover, considering the directions rightward arrow depicts a positive correlation and in contrast left arrow depicts a negative correlation. If the rightward or leftward arrows move up, it means the second variable causes the first variable which signifies a one-way causality. Further, if the rightward or leftward arrows move downwards, it depicts that the first variable causes the second variable which also signifies a one-way causality. The arrows' movement up and down considering the path of right and left, signifies a bidirectional causality between the respective variables.

As shown in Fig 2, there is a positive correlation between $CO_2$ and PGDP in the short term (Scale ranging from 0–16), and there is a bi-directional relationship in 1995, 1999, 2002, 2008, 2013, 2015 and 2018 at high frequency with a significance of 5%.

In the years 1997 and 2009, $CO_2$ emissions have a unidirectional causality to PGDP. In contrast, the years 1998, 2004, 2007, 2010, 2012, and 2014 show that PGDP had caused positive to $CO_2$ emission at 5% significance with high frequency. The period ranging from 1996 to 2019 shows a positive correlation between $CO_2$ and PGDP in the medium term (Scale ranging from 16–256). Further from 1995 to 1998, 2001, 2003 to 2007, and 2011 to 2012 there has been a bidirectional causality between $CO_2$ and PGDP with high and medium frequency at 5% significance. Thus, from 1999 to 2000, 2002, 2008 to 2010, 2013 to 2014 and in the last 3 years, which is 2018 to 2020 there has been a unidirectional causality where PGDP has caused positive $CO_2$ emissions. In the long term (Scale from 256–1024), it is visible that, from 1998 to 1999 there has been a unidirectional causality in which PGDP has caused $CO_2$ emission and from 2000 to 2010 there has been a bidirectional causality between the variables at high frequency at 5% significance. The study done using Granger found that there is a unidirectional relationship between PGDP and $CO_2$ emission. When analysing in the global context there is no causality in underdeveloped countries. Additionally, a bidirectional relationship is found between CO2 emission and PGDP in economies in transition countries for the period of 1990 to 2019 [58, 59]. Representing ASEAN countries, Malaysia, Philippines, Singapore and Thailand consist of high income, lower middle income and upper middle income countries representing that there is a unidirectional relationship between CO2 and PGDP [60].

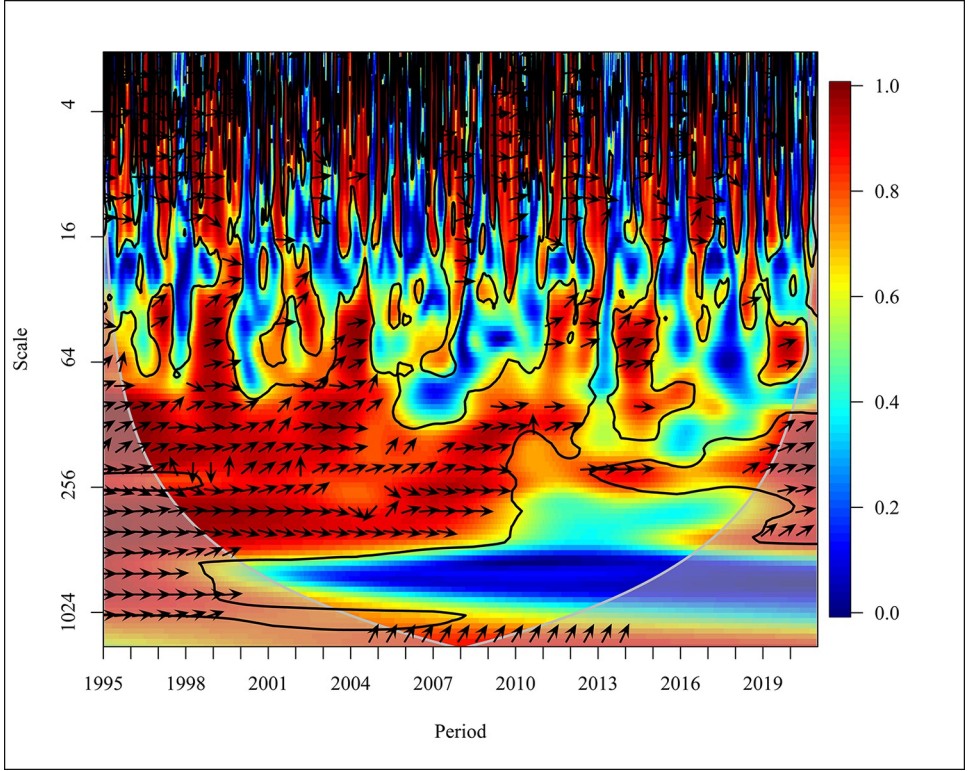

**Fig 2. Wavelet coherence: $CO_2$ vs PGDP for world.**

High-income level countries in the short term (Scale ranging from 0–16), have a mixed relationship between PGDP and $CO_2$ with high and medium frequencies, signifying that there's a positive relationship in the years 2007, 2010, 2013, 2016, and 2020 and in contrast, a negative relationship in 1999 and 2005 as shown in Fig 3. There is a cross-correlation or a bi-directional relationship between these two variables in 1999 to 2000, 2005, the second half of 2006 to 2007, 2013, and 2020 for high frequency. In the 1st half of 2006 and 2011, PGDP caused $CO_2$ with a negative impact with high frequency. In 2010 and 2016 PGDP positively caused $CO_2$ with high frequency. Further $CO_2$ has caused positive to PGDP in the second half of 2011. In the medium term (Scale from 2016–64), PGDP negatively led to $CO_2$ from 2000 to 2001 and 2007 to 2008 at medium frequency. Further in 2012 and 2013, PGDP led to $CO_2$ emissions positively. In the long term (Scale 64–256) effects are only shown in 1998 to 1999 where PGDP has positively caused $CO_2$ emission at a medium frequency. A study conducted in the USA shows that the causal flow is stronger around the period of 1910 to 2014. The study has found a bidirectional relationship between $CO_2$ and PGDP.

Nonetheless, the causality has a strong, positive long-run relationship between these two variables under the suggestion that the deterioration of the environment is due to economic expansion. In contrast, in the early 1990s, there was a reverse causality. Some factors for the rise in $CO_2$ that has been attributed can be industrialisation, energy price increments, and new technologies, while this can be disrupted by volatilities like the great depression and World War [61].

In Fig 4, Lower income classified countries have an overall positive relationship between $CO_2$ and PGDP in high, and medium frequencies in all three periods: short-term, medium-term, and long-term. In the short run (Scale 0–8), bidirectional causalities can be seen in 1995 to 1996 and 2008. In 2007, the first half of 2013, 2015, and 2018 PGDP caused positive $CO_2$

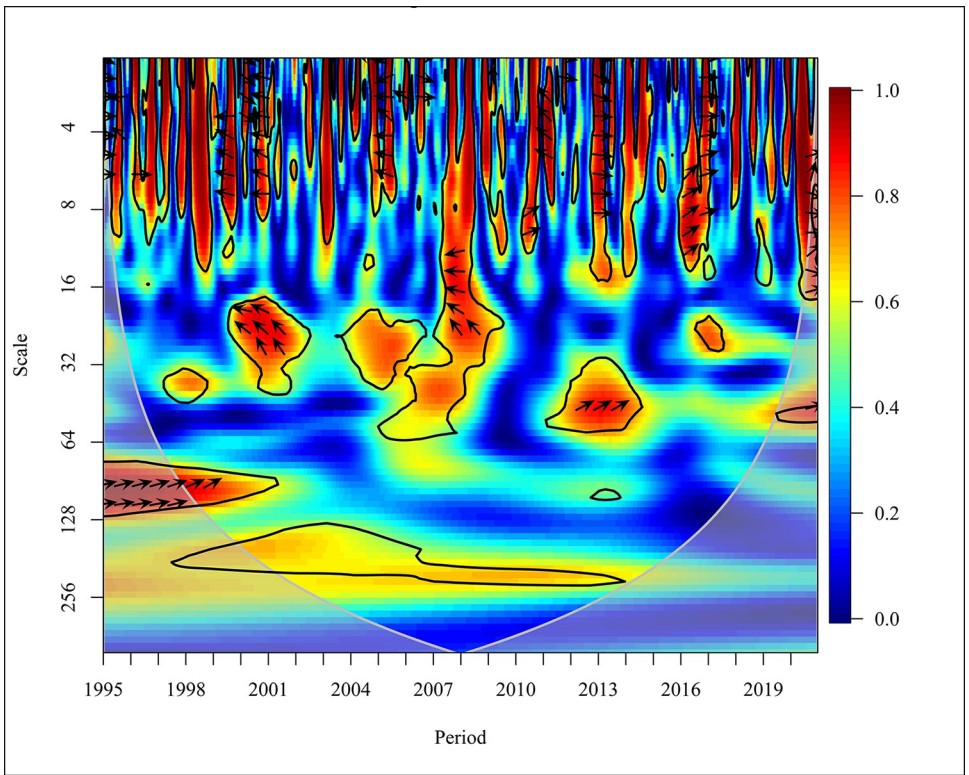

**Fig 3. Wavelet coherence: $CO_2$ vs PGDP for high-income countries.**

emissions at high and medium frequencies. Also, $CO_2$ emission has positively led to PGDP in 2004 to 2005, 2009, 2nd half of 2011, and 2019 at high and medium frequencies. In the medium term (Scale 8–32) there is a bidirectional causality between $CO_2$ emission and PGDP in the years 1996 to 1997, 2005, 2007, and 2015. Years 2004, 2006, and 2008 to 2010 there is a positive causality where $CO_2$ has led to PGDP. Only in 2017, there is a negative relationship with a unidirectional causality from PGDP to $CO_2$ is visible. From 1998 to 2018 in the long-term (32–128) has a positive relationship between $CO_2$ and PGDP in high frequency.

From 1998 to 2005, there has been a unidirectional causality where $CO_2$ has led to PGDP while from 2006 to 2017 there is a unidirectional relationship in which PGDP has led to $CO_2$. Again in 2018, $CO_2$ has led to PGDP. A study related to top oil-generating countries has categorised four countries named Congo, Guinea-Bissau, Niger, and Sudan as low-income countries which is in line with the current study. $CO_2$ and PGDP have shown a significant causal relationship in the short run with a positive one-way causality with $CO_2$ leading PGDP for the period 2000 to 2019. In the medium term, variables have a cyclic relationship and lastly in contrast to the current study in the long run from 2013 to 2019, there is a negative correlation between PGDP leading $CO_2$ [62].

For the lower-middle-income countries, Fig 5 represents a mixed relationship consisting of both positive and negative for different periods. In the short run (Scale 0–16), there is a bidirectional relationship from 1996,1998,2002,2006 to 2009, 2011, 2014 to 2017 and 2020 with high frequency. In 1999, 2004, 2012 to 2013 PGDP caused $CO_2$ emission negatively at a high frequency.

In 1997, PGDP caused $CO_2$ emission but in 2000, $CO_2$ caused PGDP positively with high frequency. In the medium term (Scale 16–64), a bidirectional relationship is depicted

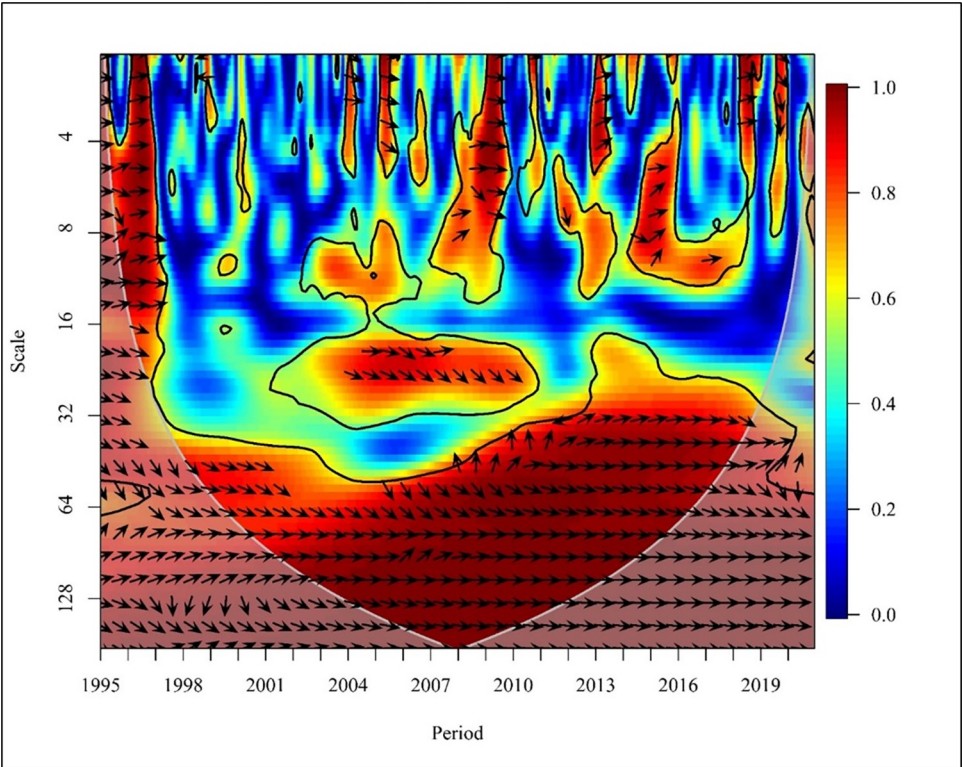

**Fig 4. Wavelet coherence: CO$_2$ vs PGDP for low-income countries.**

positively in 1996 to 1998 and 2$^{nd}$ half of 2004 to 2006 at high frequency. A unidirectional causality is depicted in 2002, 2007 2008, 2010, and 2015 which CO$_2$ has positive causality to PGDP. Only in 1$^{st}$ half of 2004, a Positive relationship with the causality of PGDP leading to CO$_2$ is depicted. In the long term (Scale 64–256) from 2000 to 2012, there is a clear positive bidirectional causality. In the 1998 to 1999 and 2013 to 2016 periods, unidirectional causality is depicted positively at high frequency. A study done in Algeria which is a lower-middle-income country has shown a short-term relationship between PDGP and CO$_2$, from 1994 to 2011. The period from 1994 to 2006, has conveyed a positive correlation with PGDP leading, and from 2006 to 2011 CO$_2$ led PGDP. The study has interpreted that there is a significant movement between PGDP and CO$_2$ in all the time scales with PGDP leading to CO$_2$. Since Algeria is an oil-rich nation and mainly relies on fossil fuels, energy consumption has driven up and has impacted CO$_2$ because when the economy grows, the carbon emissions have been expanded [33].

In upper middle-income countries in the short run (Scale 0–16), there is a bi-directional relationship in 1996, 2$^{nd}$ half of 2000, 2$^{nd}$ half of 2000, 2012, and 2018 shown in Fig 6.

A positive relation with the causality of PGDP leading to CO$_2$ emission has been shown in 2000 and 2005. In 2$^{nd}$ half of 2002 and 1st half of 2011 depict a unidirectional relationship where CO$_2$ has led to an increase in CO$_2$. Further in 1$^{st}$ half of 2002 and 2016, there has been a negative relationship where PGDP has led to CO$_2$. In the medium term (Scale 16–64), a high frequency is depicted where PGDP has led to CO$_2$ from 2004 to 2007. In the years 2011 to 2012, there was a bidirectional relationship and in 2013 there was a unidirectional relationship where CO$_2$ has led to PGDP. Long-term (Scale 64–256) depicts a clear visual of PGDP causing CO$_2$ emission at a stretch from 2003 to 2016 at high frequency with a positive relationship.

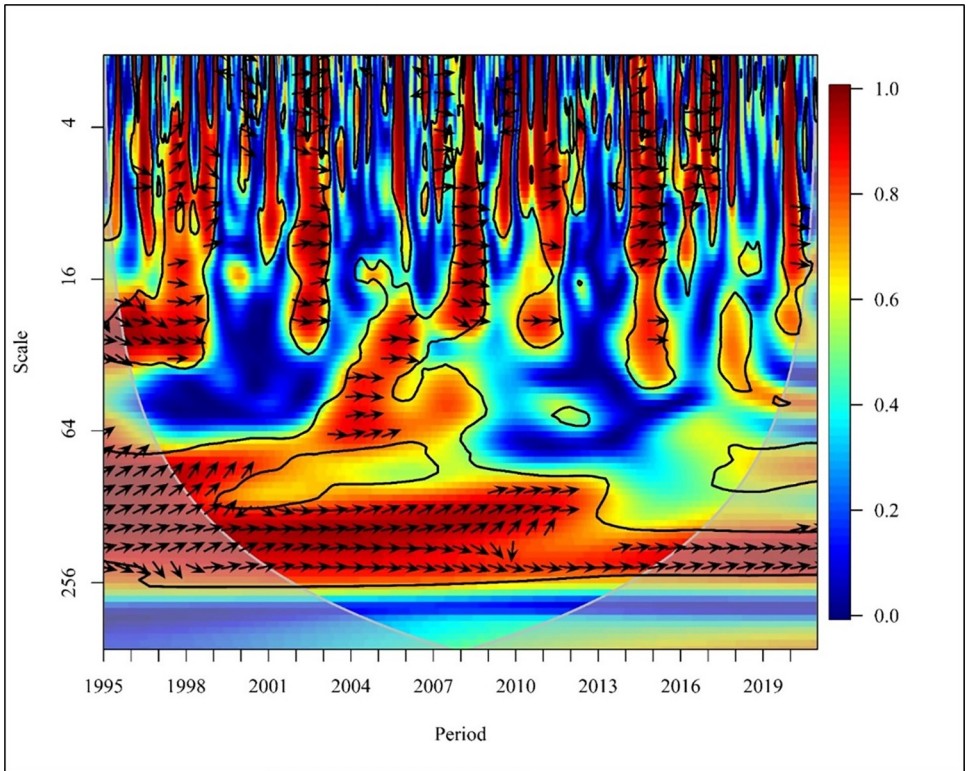

**Fig 5. Wavelet coherence: CO$_2$ vs PGDP for lower middle-income countries.**

This has indicated that economic growth has been a crucial factor in Thailand. These outcomes have depicted that Thailand should concern about the rules implemented that impact growth [36].

Fig 7 visualises the causality between CO$_2$ emissions and REC consumption in a global context. In the short term (Scale 0–16) the years 1998,1999,2004,2011 and 2015 to 2019 depict a bidirectional relationship between CO$_2$ emissions and consumption of REC at high and medium frequencies. In 1996 there was a mixed relationship among the two variables with no causality.

A study conducted to determine the bidirectional causality between energy consumption and carbon emission explains that Ghana's fuel energy consumption has resulted in CO$_2$ emissions. Therefore, the energy penetration has not reached the level required to start the reduction of CO$_2$ emissions as confirmed by a study [37]. Further, the years 1995 and 2007 depict that consumption of REC caused a reduction in CO$_2$ emission while year's 2010. Exceptionally, 2012 depicts that CO$_2$ emission causes a negative impact on the consumption of REC at high and medium frequencies. In the medium term, globally (Scale 16–256), a bidirectional causality is depicted in the years 2002, 2007 to 2010, and 2015 to 2016 at high frequencies. Further, unidirectional causality is depicted in the years 2001, 2004, 2014, and 2018 in which REC has caused a negative impact on CO$_2$ emission. The years 1998 and 1999 show a unidirectional relationship where CO$_2$ negatively impacts REC. In the long term (Scale 256–1024) shows low frequency with no relationship between the variables in a global context. In contrast, a global study conducted based on 138 countries supports a bidirectional energy emission nexus between REC and CO$_2$ Emission. Research shows REC increases carbon emissions, while capital market investment boosts REC [63].

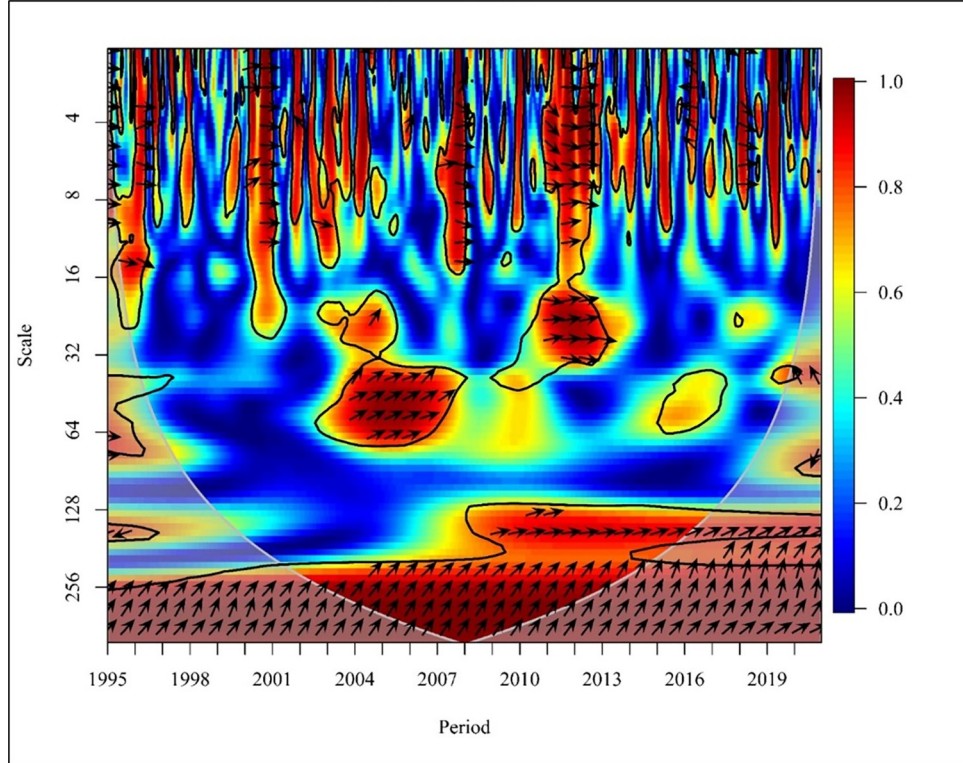

**Fig 6. Wavelet coherence: CO₂ vs PGDP for upper middle-income countries.**

Unlike in the global context, Fig 8 shows high-income country category has a mixed relationship between $CO_2$ and REC in the short term (Scale 0–16) at high and medium frequencies. A bidirectional causality is depicted in most years while a unidirectional causality is depicted in 1999 when $CO_2$ emissions caused negatively on REC at 5% significance.

Stepping away from normality, towards the latter part of 2016, REC has positively caused $CO_2$ emissions. In the medium term (Scale 16–64), a bidirectional causality is depicted in the latter half of 2007 till the end of 2008 and 2012 at high frequency while the unidirectional relationship is shown in 2002, and 2005 at high and medium frequency with 5% significance. In the long term (Scale 64–256) unlike in the global context which did not show any relationship, the high-income category depicts a unidirectional relationship in the years 1997 and 1998. In relation to the study conducted in Japan reveals wavelet coherence for $CO_2$ and REC. Between 1993 and 1997, the series was out of sync with $CO_2$ emissions [64]. Also, 2009 to 2010 the measures act in phase with $CO_2$ and REC which aligns with the findings of the study. Overall, the findings reveal that higher usage of REC lowers $CO_2$ emissions. Thus, investment in RE is environmentally beneficial.

In Fig 9, for the low-income category, in the short term (Scale 0–8) the beginning of 1999 shows a bidirectional relationship while at the end of the year and 2004 a unidirectional relationship is shown which REC causes $CO_2$ emission. 2000 and 2007 years show a unidirectional relationship in which $CO_2$ negatively impacts REC with high frequency at 5% significance.

The year 2008 shows a different behaviour of the variable which is that $CO_2$ has caused a positive impact t to REC at high frequency. Considering the medium term (Scale 8–32) which shows a negative relationship between $CO_2$ and REC at high and medium frequency, a bidirectional causality is visible from 1996 to 2000 and in 2009 and 2010. Further, a unidirectional

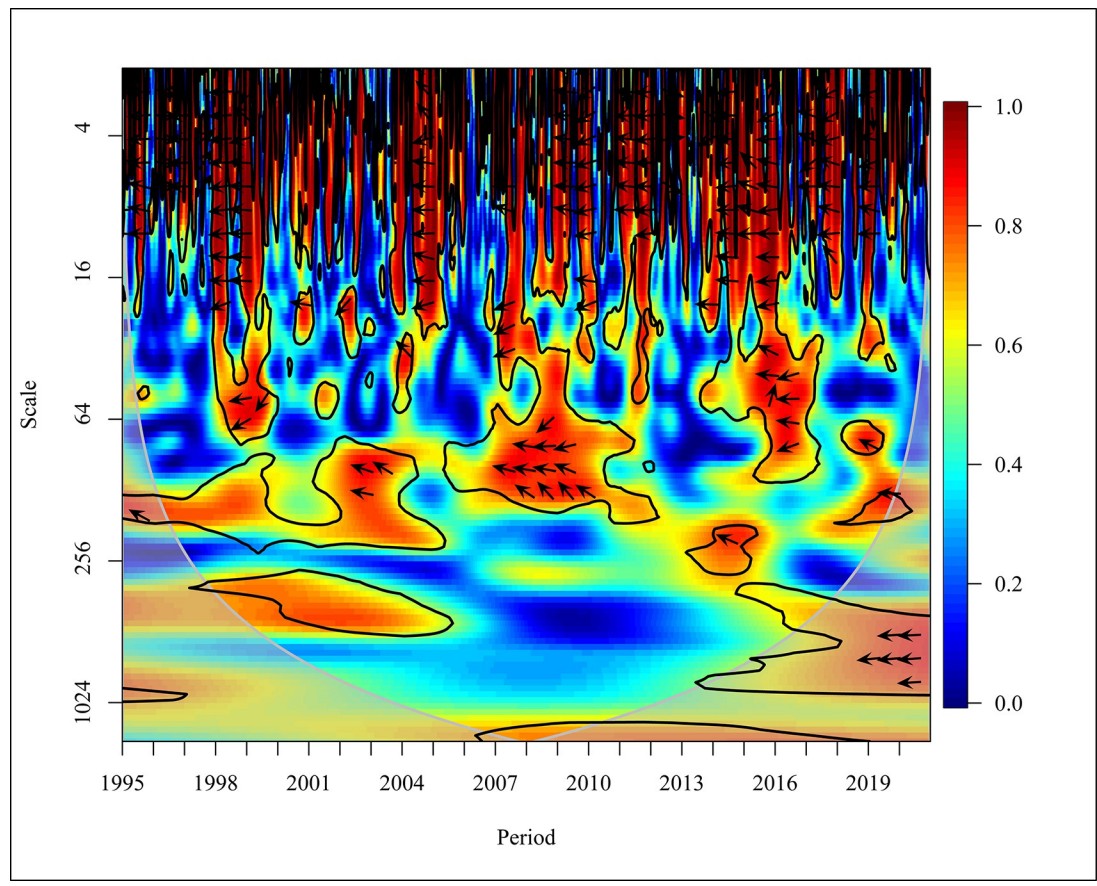

**Fig 7. Wavelet coherence: $CO_2$ vs REC for world.**

relationship is depicted in 2011 where REC caused $CO_2$, and years 2007 to 2008, $CO_2$ has led to REC negatively at a 5% significance level. Exceptional behaviour is visible where a unidirectional positive causality from $CO_2$ to REC is depicted in the years 2012 and 2015. High and medium frequencies are depicted in the years 1998 to 2017 with a negative relationship whereas a bidirectional causality is shown from 2006 to 2015 in the long term (Scale 32–128) at 5% significance. Unlike in the global scale or high-income country category, the low-income country category shows a unidirectional relationship in 1999, 2000, 2004, and 2005 where $CO_2$ causes REC at a high frequency. Furthermore, a clear bidirectional causality between $CO_2$ and REC is visible from 2006 to 2015.

Fig 10 visualises the causality in lower-middle-income countries from 1995 to 2020 in the short term (Scale 0–16) as a bidirectional relationship between $CO_2$ and REC which is commonly shown throughout the period, except for 1997, 2001, 2009, and 2012 to 2016 which visualises unidirectional relationships at high frequency with 5% significance.

The year 2001 shows a one-way causality in which $CO_2$ impacts REC negatively while in 2009 and 2016 REC causes $CO_2$ negatively. Further, in 1997 there was a unidirectional positive causality in which REC caused $CO_2$ and in 2012 $CO_2$ positively caused REC at 5% significance. In the medium term (Scale 16–64), only the low middle-income category exhibits no bidirectional causation when compared to the medium term of other country categories. Unidirectional causality is visible in the years 2007, 2014, and 2020 where REC causes negative to $CO_2$. Years 2000, 2008, and 2013 show a unidirectional relationship where $CO_2$ negatively causes

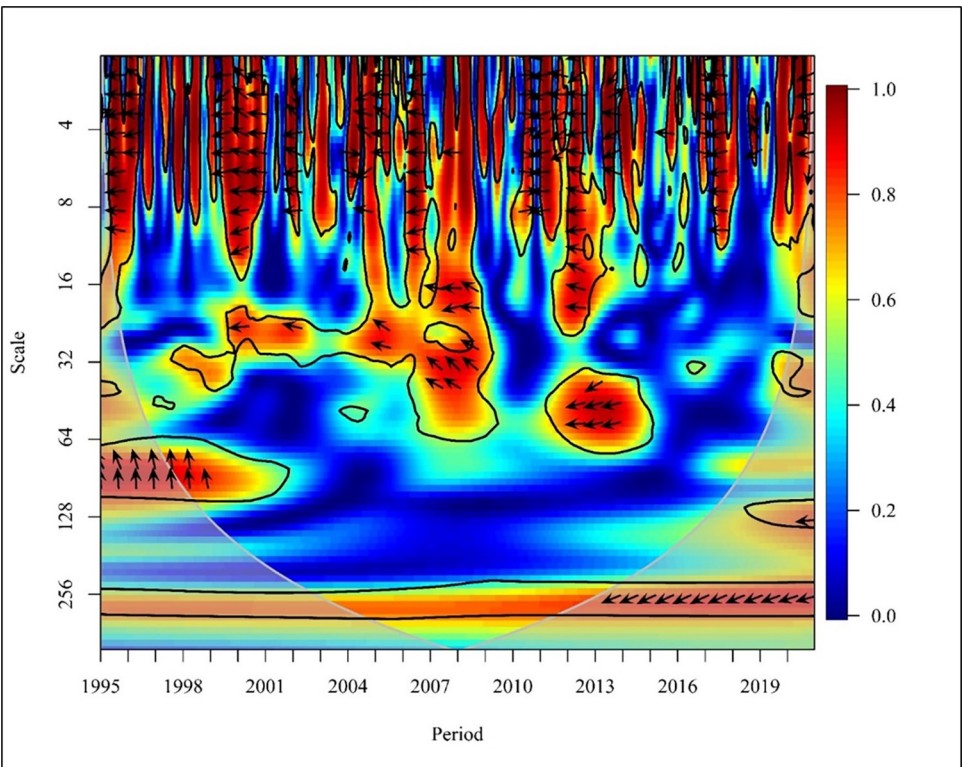

**Fig 8. Wavelet coherence: $CO_2$ vs REC for high-income countries.**

REC at high frequency with 5% significance. In the long term (Scale 64–256) from 2000 to 2018, there is a negative relationship between the $CO_2$ and REC where the years 2004 to 2018 specifically show a bidirectional causality between the two variables at a high and medium frequency at 5% significance. Further, a unidirectional relationship is indicated where $CO_2$ causes negative REC from 2000 to 2003.

In Fig 11, the upper middle-income country category, in the short term (Scale 0–16), bidirectional relationship is depicted in 1995, 1999 to 2004, 2007 to 2008, 2013 to 2014, 2016, and 2019 at a high and low frequency at 5% significance level.

In the years 1996, 2005, 2012 and 2015 there is a unidirectional relationship where $CO_2$ impacts negatively to REC. Examining the positive correlation further, we find that, in 2006 and 2018, $CO_2$ positively caused REC, while in 2011, REC positively caused $CO_2$. In the middle term (Scale 16–64) in upper middle-income countries, bidirectional relationships are depicted in 1997 to 2001 at high frequency, while unidirectional relationships are visible in 2004, 2009, and 2014 where $CO_2$ has caused REC. Further, in the years 1997 and 2017, consumption of REC has negatively impacted $CO_2$ emissions. In the long term (Scale 64–256), there is a negative relationship from 1998 to 2001 and 2013 to 2018 in high and medium frequency. The years 1998 to 2001 and 2014 to 2016 depict a unidirectional relationship where REC has caused $CO_2$ emission negatively at a 5% significance level. A study on the impact of REC, PGDP, and net exports on consumer-based $CO_2$ emissions shows that Dynamic Ordinary Least Square's estimation of REC lowers $CO_2$ emissions over time with an increase of 1% in REC and a decrease of 0.26% in $CO_2$ emissions. On the other hand, empirical results show that PGDP and imports have a beneficial impact on $CO_2$ emissions, which is a 1% rise in PGDP increasing by 0.46% of $CO_2$ [65].

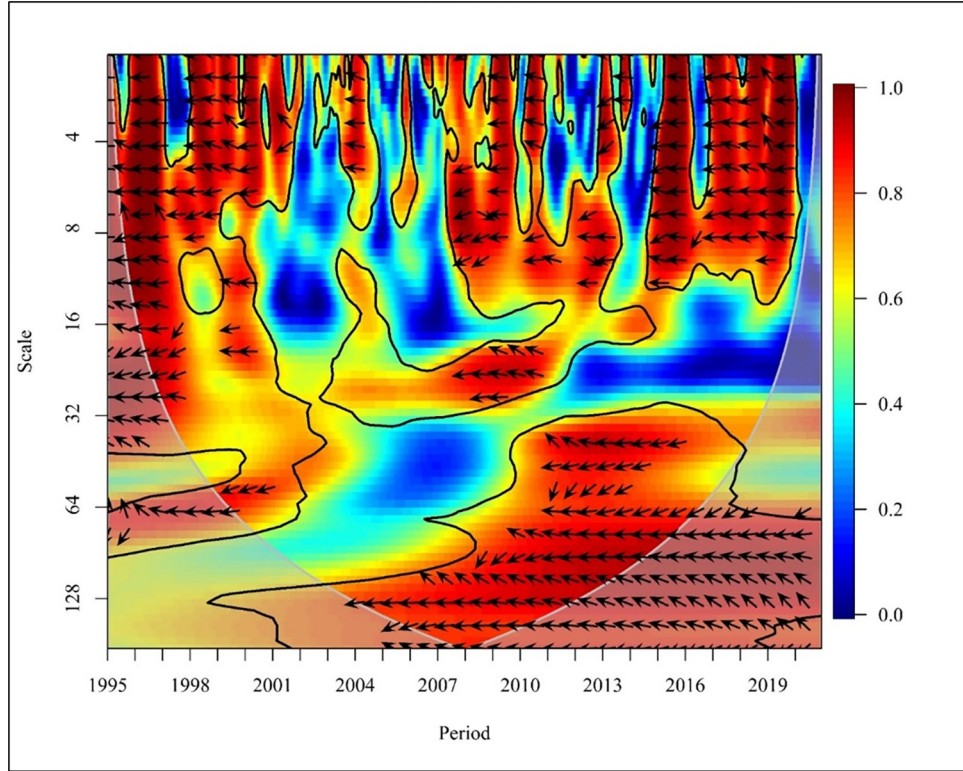

**Fig 9. Wavelet coherence: $CO_2$ vs REC for low-income countries.**

The Usage of non-renewable energy like coal, natural gases, and petroleum has increased significantly over the last four decades in the period from 1980 to 2016 [66]. In the global context, a bidirectional causality was depicted negatively in 1995, 1998 to 1999, 2005, and from 2016 to 2019 with high frequency at 5% significance as shown in Fig 12. An empirical study which was done using data observed from 65 countries revealed a bidirectional causality in the short and long term. The causation for the results of a bidirectional causality is due to the industrial and economic growth in the world [39]. In the years 2007, 2009, and 2015, a unidirectional causality was depicted where $CO_2$ causes NREC. In 2010, NREC became the major cause of $CO_2$ emission. The variations of NREC and $CO_2$ in the medium term (16–64) in the years 1997 and 2018 show a positive relationship and a bidirectional causality was identified from 2002 to 2004, 2007 to 2009, and 2016 to 2017.

In the years 1998 to 1999, a unidirectional causality was identified where NREC led to $CO_2$ emission at a 5% significance level. A comparative study done in Thailand and BRICS countries has supported the result that an increase in consumption of non-renewable energy has led to an increase in $CO_2$ emission [50]. Furthermore, a unidirectional causality is depicted in 2001, 2014, and 2019 where $CO_2$ has caused positively on NREC with high and medium frequencies at a 5% significance level.

Fig 13 depicts, in high-income countries, a short term (Scale 0–16) and bidirectional causality between $CO_2$ emission and NREC for the period except for 1999 and the 1$^{st}$ half of 2004 which shows a unidirectional relationship where NREC has a positive cause to $CO_2$ emissions at 5% significance level. $CO_2$ emission has shown a negative and significant result which depicts that increases in $CO_2$ emission have contributed to a reduction in the use of non-renewable energy in the short run [67]. In the medium term (Scale from 16–64), a positive relationship at high significance is seen from 2000 to 2008 and 2012 to 2013.

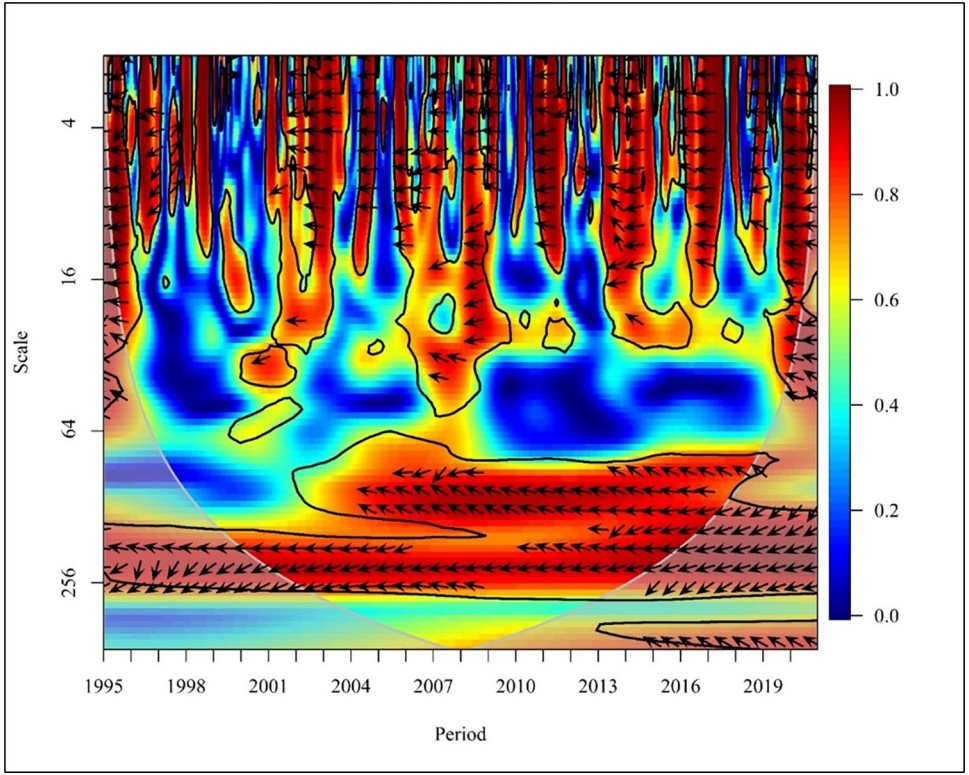

**Fig 10. Wavelet coherence: $CO_2$ vs REC for lower middle-income countries.**

A bidirectional causality is visible from 2007 to 2008 and 2012 to 2013. Unidirectional relationships are visible in 2000 where NREC led to $CO_2$ emission while in the years 2002 and 2005, $CO_2$ led to an increase in NREC. Considering the long term (Scale 64–257) from 1998 to 1999, a unidirectional relationship is visible where $CO_2$ has led to an increase in NREC. The rest of the period shows low frequency.

In low-income countries from 1995 to 2019, a positive relationship is shown between $CO_2$ and NREC with mixed frequencies in the short term (Scale 0–8) which is illustrated in Fig 14. In the years 1999, 2001, 2015, and from 2017 to 2019, a bidirectional relationship is depicted at high frequencies. Unidirectional relationships where NREC has led to $CO_2$ emission are visible in the years 1998, 2000, 2008 to 2009, and 2012 to 2013.

The years 1995 to 1997, 2004, and 2010 depict unidirectional causality where $CO_2$ had led to an increase in NREC at high frequency at a 5% significance level. Results of medium term (Scale from 8–32) from 1997 to 2000 and from 2009 to 2010 show a bidirectional causality with high frequency. NREC has led to $CO_2$ emissions showing a unidirectional causality in 2006, 2008 and 2013 while $CO_2$ has led to an increase in NREC in 2004 to 2011, and 2015. In the long term (Scale 32–128) from 2008 to 2013, a bidirectional causality is shown while in 1999 to 2000 and 2014 to 2016, a unidirectional causality is depicted where NREC leads to increase $CO_2$. Also, from 2006 to 2007, $CO_2$ has led positively to NREC.

In Fig 15, the Lower middle-income countries in the short-run (0–16) show a bidirectional causality in the years 1996,1997, from 2002 to 2006, 2008, 2011, and 2013 at a high frequency. There is a bidirectional causality among $CO_2$ emission and NREC with a 95% confidence level [25]. In 2013, there was a significant hike in NREC-led CO2 emissions compared to 2004 and 2011.

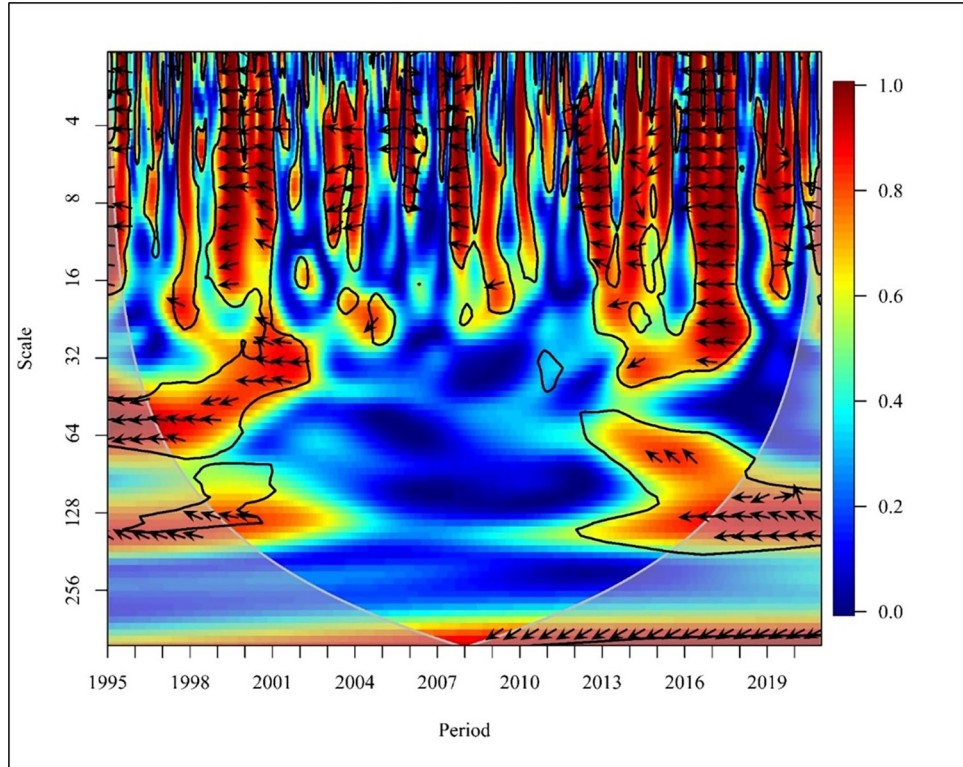

**Fig 11. Wavelet coherence: $CO_2$ vs REC for upper middle-income countries.**

In the long term (64–256), a bidirectional causality has been identified from 2004 to 2016. A unidirectional relationship in which NREC Causes $CO_2$ is visible in 1995, 1997, 2001, 2007, and 2014. $CO_2$ has caused positive effects on NREC in the years 1997, 2009, and from 2015 to 2016. Exceptionally in the years 1996 and 2012, a negative relationship where NREC led to a decrease in $CO_2$ emission is depicted. In long term (Scale 64–256) shows bidirectional causality between NREC and $CO_2$ at high frequency from 2004 to 2016. It is proven by the evidence generated by the results that there is bidirectional causality between $CO_2$ emission and NREC [39]. From 2001 to 2003, there has been a unidirectional causality where NREC positively caused $CO_2$.

In the upper middle-income countries years 1995,2001, 2003 to 2004, 2007 to 2008, 2013 to 2014, and 2019 in short-term (0–16) resulted in a bidirectional causality between $CO_2$ emission and NREC at high frequency in Fig 16. 1997, 1999 to 2000, 2005, and 2015 to 2016 show right downward arrows revealing NREC has affected positively $CO_2$ emission at a 5% significance level.

In exception to normal behaviour, 2005, 2011, and 2012 a unidirectional negative causality is shown at high frequency. In the medium term (Scale 16–64) from 1997 to 1999 show a bidirectional relationship. The future unidirectional relationship of NREC causing $CO_2$ is depicted in 2004, 2009, and 2013 to 2014. Also, $CO_2$ positively led to NREC from 2001 to 2002 and 2017 at a 5% significance level. In the long run (64–256) the results show the right downward arrow in 1998 and 2015 to 2016 with high and low frequencies revealing that $CO_2$ emission has caused NREC. A similar study reveals that there is a unidirectional causality running from $CO_2$ to NREC in EU countries [67].

In this study, the causality will be analysed for three periods: short-term, medium-term, and long-term from 1995 to 2020 with different frequencies. A summarisation of the

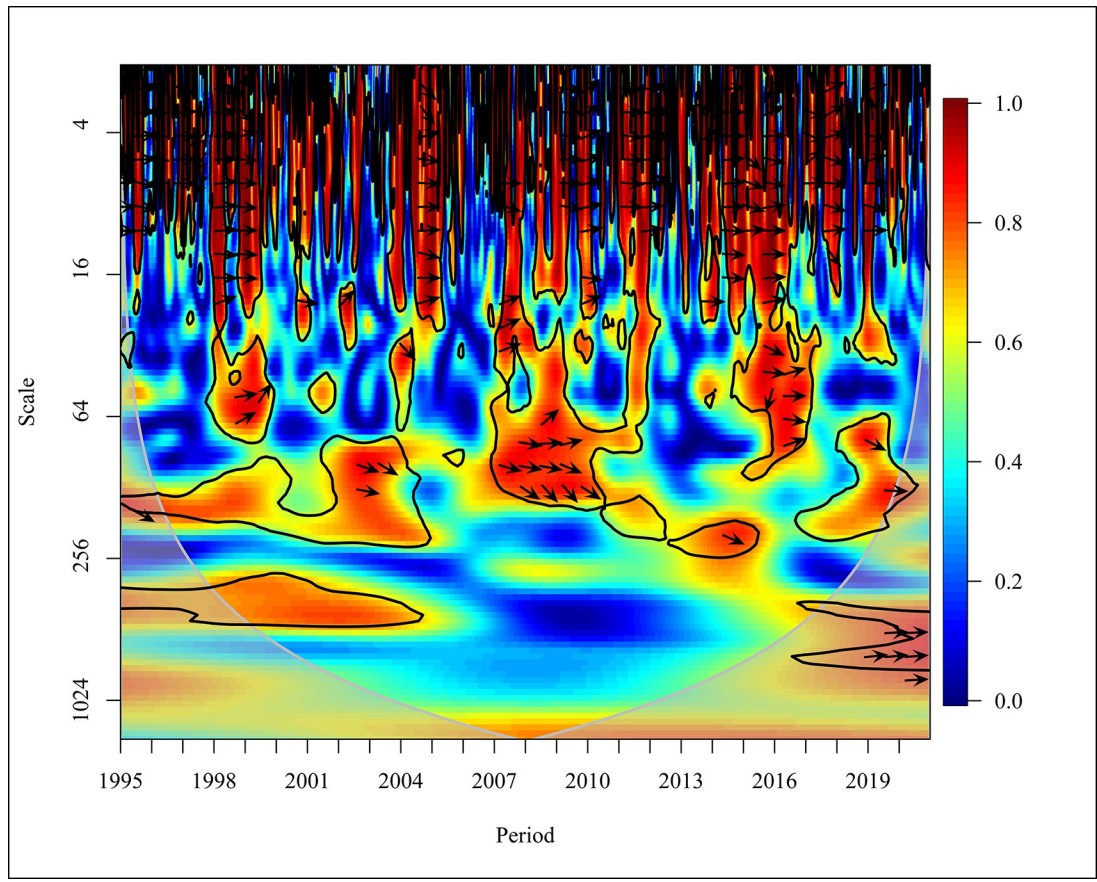

**Fig 12. Wavelet coherence: CO$_2$ vs NREC for world.**

correlation and causality of the variables for the aforementioned time period classified by five years is shown in Fig 17.

The results of the wavelet coherence indicate the existence of a significant correlation between PG DP and CO$_2$ emission and energy consumption and CO$_2$ emission in all the time scales. In short term, medium term and long term, PGDP has both bidirectional relationship and unidirectional relationships in a global context which is similar to the study based on 30 countries which has shown that the countries that depend on fossil fuels have had economic growth while increasing the environmental deterioration like CO$_2$ emission with a mixed relationship while centering for bi-directional causality [68]. But in short term there is a one-way causality between PGDP and CO2 emission except for the 2006 to 2010 time range.

Further, all the country categories except for high income and low middle income have a unidirectional causality. Focusing on medium term, high income shows a one-way causality throughout the time period. In the long term, all the categories have one-way causality except for the upper middle-income category. REC also has a mixed causality with both bi-directional relationships and unidirectional relationships. In short, there is a negative relationship with bidirectional causality throughout the time range [69, 70] and in contrast, [71] it shows no causality globally and region-wise. In the medium term, there is a one-way causality from 2006 to 2020 in upper-middle-income and lower-middle-income categories in the first and the last five years of the given time period.

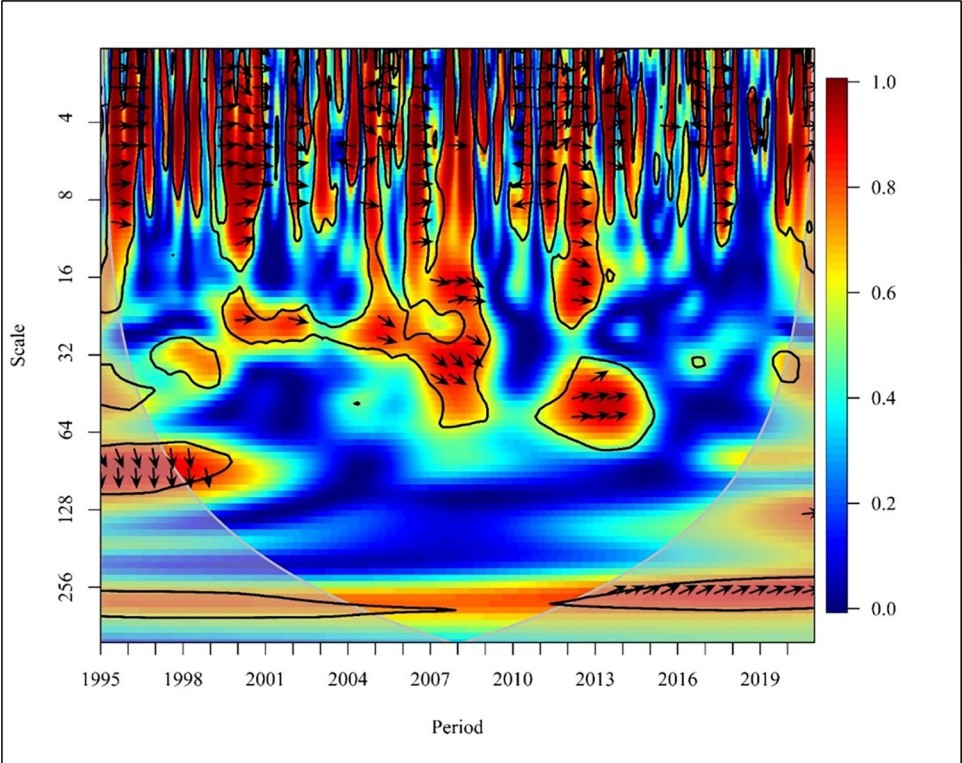

**Fig 13. Wavelet coherence: $CO_2$ vs NREC for high-income countries.**

In the long term, there is no causality shown globally. One causality can be seen in the first period of high income and throughout the upper middle income with no causality from 2006 to 2010. Further, a mixed causality can be seen in lower income and lower middle-income categories. Similarly, NREC also has both bidirectional and unidirectional relationships. In the short term, there is a bidirectional relationship globally except for the years 2006 to 2015. In the medium term similar to REC there is a one-way causality in upper middle income, and it shows a right upward behaviour which explains NREC causes $CO_2$ in upper middle income from 2006 to 2015 and a downward arrow, where CO2 causes NREC in lower middle income from 2011 to 2020. Moreover, in comparison to the REC directional behaviour there is a large number of one-way causalities. Further, in the long term, there is a one-way causality throughout in upper middle income.

The Granger causality test results obtained for each country under each income category are shown in S2–S4 Appendices. Before the Granger causality test, the unit root test was applied to ascertain the stationarity of the variables. Considering the relationship between $CO_2$ emission and PGDP, RE, and NRE in general there is a reasonable confirmation that Granger causes each other in all income categories (Table 5).

Cross-country results of Granger causality between $CO_2$ and PGDP are presented in S2 Appendix. The results show that under the high-income country category, the countries Finland, Poland, Romania, and the Slovak Republic show clear bidirectional causalities between $CO_2$ and PGDP. Furthermore, 19 other countries show unidirectional behaviour where 13 countries show $CO_2$ causing GDP and 6 other countries show GDP causing $CO_2$ with the rest of the 22 countries showing no causality. Considering the upper middle-income country category, the Dominican Republic shows two-way causality while Libya, Albania, Dominica

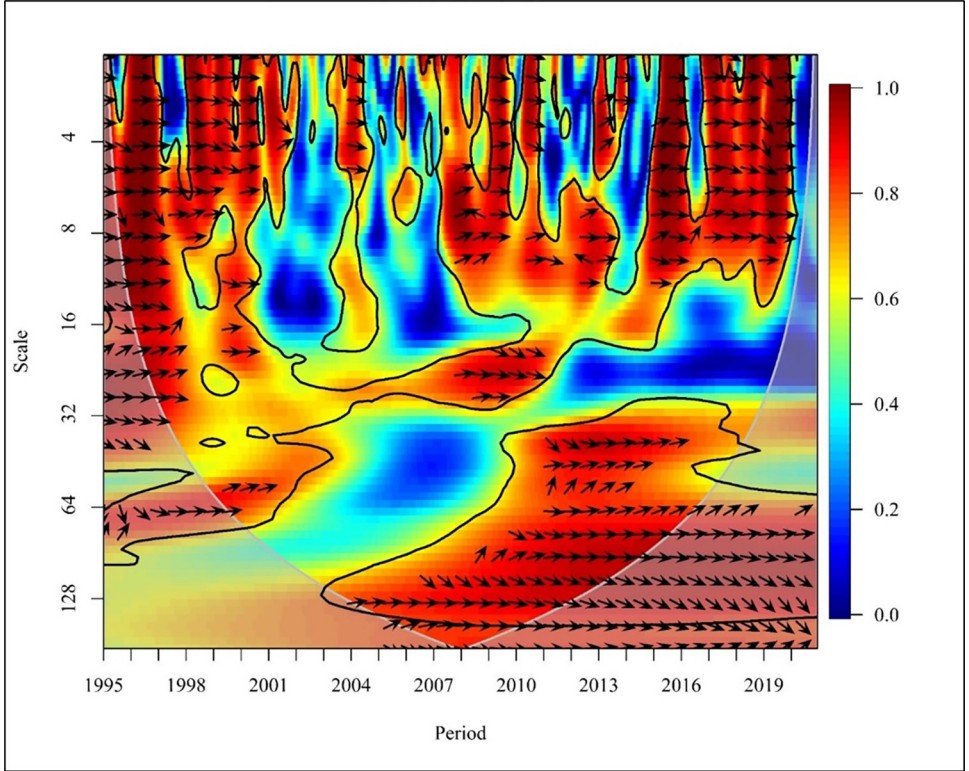

**Fig 14. Wavelet coherence: $CO_2$ vs NREC for low-income countries.**

Marshall Islands, and St. Vincent depict unidirectional causality of $CO_2$ causing PGDP, and Grenadines and Botswana, El Salvador, Gabon, Jamaica, Malaysia, Mexico depict PGDP causing $CO_2$. A study reveals a bidirectional relationship between $CO_2$ and GDP in economies in transition, a unidirectional relationship for developing countries, and no causality for developed and least developed countries, covering 152 countries, analysed through Granger causality [59].

Another study reveals that GDP growth drives $CO_2$ emission in the US, France, Australia, and Germany with a unidirectional relationship and growth in $CO_2$ emission drives that in GDP in China, India, Brazil and Japan [72].

Additionally, low-middle-income countries and low-income countries show unidirectional and no causality among the countries. In the low middle-income category Algeria, Angola, Benin, Haiti, Kiribati, Lesotho, Mongolia, Samoa, Tajikistan, Tunisia, and Bangladesh show that $CO_2$ causes GDP while Cabo Verde, Ghana, Guinea, Honduras, Nepal, Vanuatu, Bhutan, and all other 33 countries show no causality. Under the low-income country category, Togo, the Central African Republic, Chad, Niger, and the Syrian Arab Republic show unidirectional relationships with $CO_2$ causality to PGDP while Madagascar, Mali, Sudan, and Uganda show GDP Causality to $CO_2$ also all other 11 countries show no causality.

Considering all country categories, the cross-country Granger results of $CO_2$ and RE are shown in S3 Appendix. Bidirectional causalities are present in Qatar, Trinidad and Tobago, France, Portugal, Romania, and Uruguay in the High-income country category. Furthermore, unidirectional relationships with $CO_2$ causality to RE are depicted in Seychelles, Australia, Cyprus, Finland, Germany, Israel, and New Zealand while RE causality to $CO_2$ is depicted in Andorra, Canada, Iceland, Panama, Singapore, Spain, St. Kitts, and Nevis, Switzerland, and

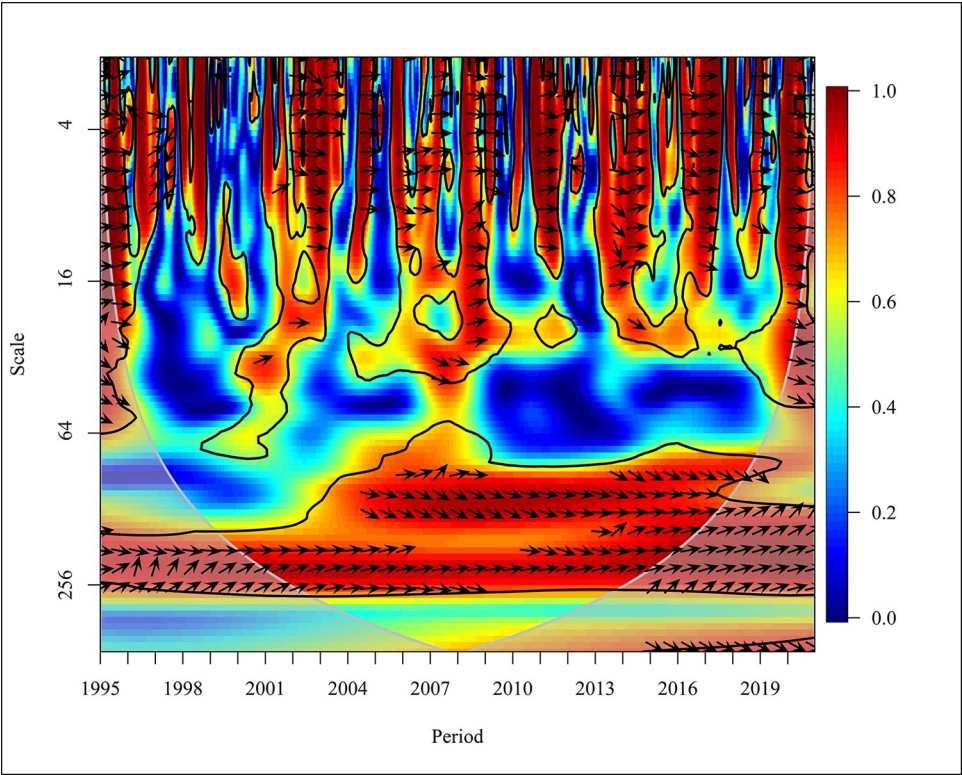

**Fig 15. Wavelet coherence: CO$_2$ vs NREC for lower middle-income countries.**

Saudi Arabia. 23 other countries show no causality. Upper middle-income country categories St. Lucia, Equatorial Guinea, and Maldives show bidirectional Granger while 11 other countries show unidirectional causality. Out of the 11 countries Belize, Guatemala, Mauritius, Jamaica, Turkey, and South Africa show CO$_2$ causes RE, while Cuba, Fiji, Costa Rica, Indonesia, and Kazakhstan show GDP causing CO$_2$.

Guinea shows a bidirectional relationship under the low-middle-income country category, with Honduras, India, and Tunisia showing unidirectional causality of CO$_2$ causing RE. In addition, Algeria, Cabo Verde, Egypt, Arab Rep., Mongolia, and Bolivia countries show that RE causes CO$_2$ emission. 42 countries show no causality under low-middle-income countries. Considering Low-income countries, the Granger between CO$_2$ and RE is bidirectional in the Central African Republic, Chad, and Congo, Dem. Rep. Unidirectional Granger with CO$_2$ causing RE is present in Mali, Rwanda, and Uganda while RE causing CO$_2$ is present in Guinea-Bissau, Niger, Sudan, and the Syrian Arab Republic. A study done to analyse the causal relationship between the same variables in the current study reveals a strong relationship among variables signifying that there is a bidirectional relationship between RE and CO$_2$ [43].

Similar results to the Granger causality between CO$_2$ emission and RE are depicted cross-country for CO$_2$ emission and NRE, which are presented in S4 Appendix.

## Conclusion and policy implications

The causality among economic growth, renewable energy, and non-renewable energy has been analysed using the wavelet coherence method for the period 1998–2020. Throughout the period in short term a bidirectional causality has been shown between CO$_2$ emission among renewable and non-renewable energy in most years while no causality has resulted from 2006–

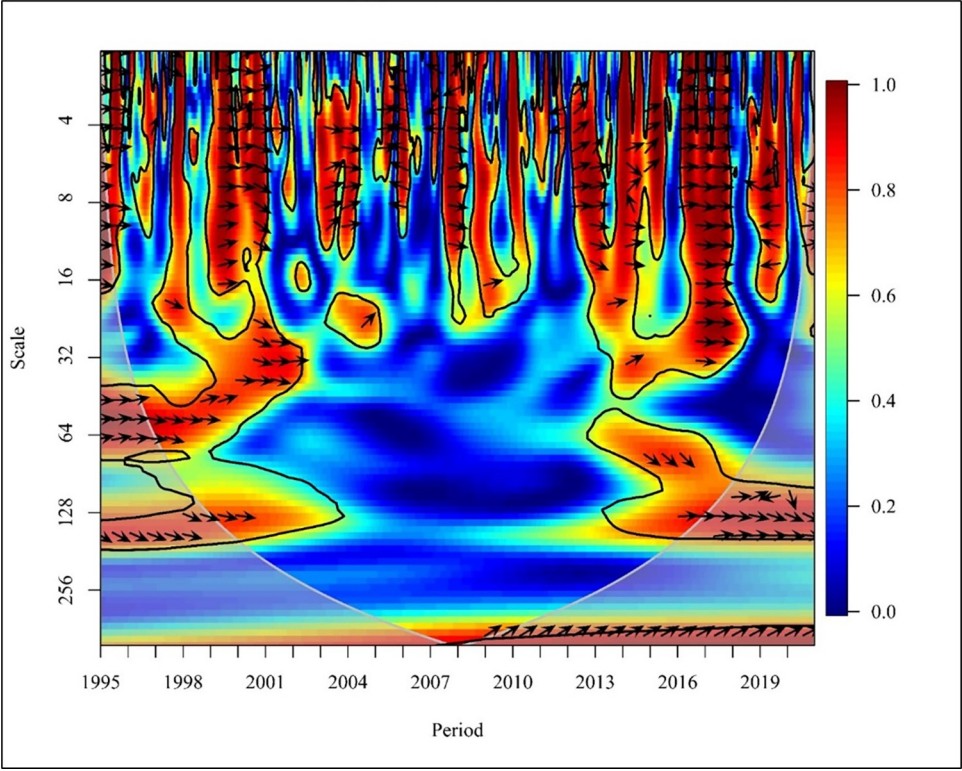

**Fig 16. Wavelet coherence: $CO_2$ vs NREC for upper middle-income countries.**

2010. In medium term, from 2016–2020, in high-income countries, from 1995 to 2000 and 2016 to 2020 in low-income countries, in the long term no causality among $CO_2$ emission and NREC has been shown from 1995–2020 and in high income countries from 2001–2020. Negative bi-directional causality was found in the world from 1995–2020 and in low-income countries in the short term while a positive relationship has resulted from 1995–2020 in low-income countries. Further to confirm the results taken by Wavelet coherence, cross-country analysis has been done using the Granger causality test which reveals mixed relationships for different countries which includes, bi-direction, uni-direction and no causality between $CO_2$ and PGDP, $CO_2$ and REC and $CO_2$ and NREC. Further comparing the two methodologies reveals all the income group categories have a bidirectional causality between all three variables mentioned above in general.

Policymakers can concentrate on initiatives like promoting renewable energy sources, directing investments towards sustainable technologies, strengthening and enforcing regulations to reduce emissions. The utilisation of renewable energy from hydro, solar, wind, biomass, and geothermal sources presents a more sustainable option, leading to reduced $CO_2$ emissions [73, 74]. Comprehending the scale and features of these $CO_2$ emissions is crucial for devising effective mitigating strategies [75–77].

Achieving net zero carbon emissions typically involves two approaches. Firstly, it involves offsetting carbon emitted into the atmosphere by investing in sustainable technologies like carbon capture and storage. Secondly, it entails averting future carbon emissions by prioritising renewable energy sources. Various organisations have launched collaborative projects to curb $CO_2$ emissions from industries such as cement and steel, which are recognised as major contributors to $CO_2$ emissions in production economies. Many studies have shown that switching

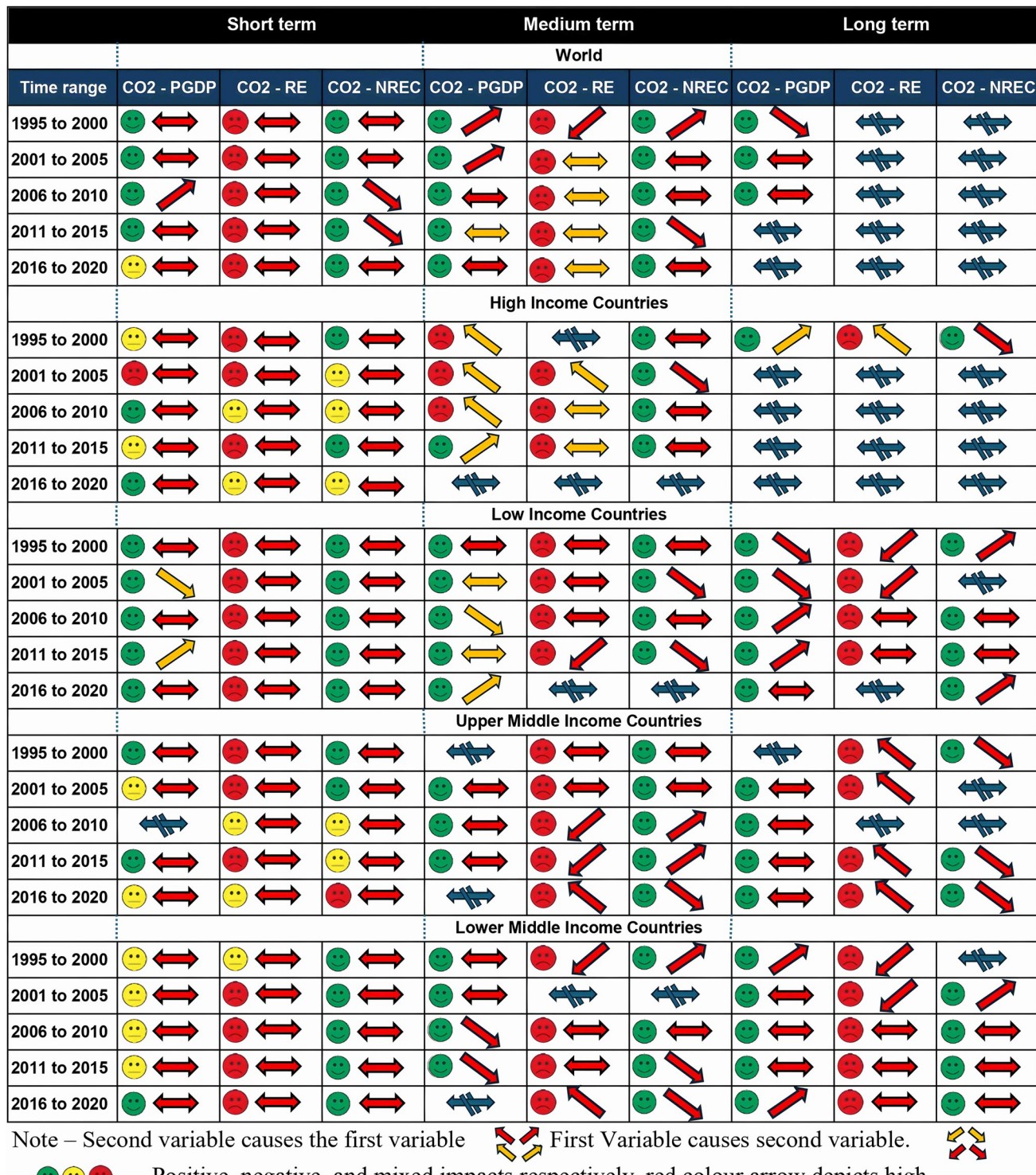

**Fig 17. Summary of wavelet coherence graph.** Source: Authors' Compilation.

**Table 5. Summary findings of wavelet coherence and Granger causality between $CO_2$ and PGDP, $CO_2$ and RE, and $CO_2$ and NRE.**

| Income Category | Wavelet Coherence | Granger Causality |
|---|---|---|
| Global | Bidirectional | Bidirectional |
| High Income | Bidirectional | Bidirectional |
| Low Income | Bidirectional | Bidirectional |
| Upper Middle Income | Bidirectional | Bidirectional |
| Lower Middle Income | Bidirectional | Bidirectional |

Source: Authors' Compilation.

to renewable energy leads to lower $CO_2$ emissions. With widespread concern about greenhouse gases, it's expected that the level of $CO_2$ emissions will greatly affect how much renewable energy has been used [78–84]. Governments should take actions by implementing programs such as promoting public transportation for urban traveling and imposing additional charges for the use of personal vehicles within cities.

By effectively addressing the intricate relationship between economic growth, renewable energy and non-renewable energy, our study highlights several policy recommendations supported by the results of the granger causality test and wavelet coherence approach. Promoting renewable energy through incentives such as grants, tax incentives, low interest loans to mitigate carbon emissions which fosters the economic development. Another crucial strategy is investing in sustainable technologies, including carbon capturing technologies, energy efficient systems, green hydrogen technologies and smart grids. Moreover, the causality results indicate, even though economic expansions tend to increase carbon emissions, with the implementation of these innovative technologies can mitigate this relationship. Enforcing regulatory frameworks is also a pivotal policy implication which aligns with the causality of the variables in the study. By employing stringent emissions regulations and carbon pricing mechanisms like carbon taxes, ensures that polluters internalise the costs of pollution, thereby incentivising cleaner technologies. Addressing urban transportation emissions is critical, with strategies such as electric vehicles, non-motorized transportation modes will tend to reduce emissions. Improving the urban transportation infrastructure will enhance economic efficiency while reducing pollution. Additionally, international collaborations will play a crucial role in addressing global climate change challenges. Collaborative efforts such as technological transfer agreements, fostering partnerships, participating in global climate initiatives, and developing policy frameworks, have created synergies that amplify the positive impact of individual national efforts in mitigating $CO_2$ emissions.

However, it is important to acknowledge the limitations of this study, including the unavailability of data which are relevant for the recent years. Although wavelet analysis is an effective method for analysing non-stationary signals in time-frequency space, its interpretation of correlation patterns can be distorted due to its sensitivity to wavelet basis and parameter selection. A limited number of studies have been done using the wavelet coherence method in a global context. Future researchers could incorporate additional variables to provide a more comprehensive understanding of emissions. In addition, indicators like other greenhouse gas emissions can be used for future studies as possible extensions.

## Supporting information

**S1 Appendix. Data file.**
(XLSX)

**S2 Appendix. Cross-country analysis between CO$_2$ and PGDP for all income categories using Granger causality.**
(DOCX)

**S3 Appendix. Cross-country analysis between CO$_2$ and RE for all income categories using Granger causality.**
(DOCX)

**S4 Appendix. Cross-country analysis between CO$_2$ and NRE for all income categories using Granger causality.**
(DOCX)

## Author Contributions

**Conceptualization:** Yuganthi Caldera, Tharulee Ranthilake, Ruwan Jayathilaka.

**Data curation:** Yuganthi Caldera, Tharulee Ranthilake, Heshan Gunawardana, Dilshani Senevirathna, Ruwan Jayathilaka.

**Formal analysis:** Yuganthi Caldera, Tharulee Ranthilake, Ruwan Jayathilaka.

**Investigation:** Yuganthi Caldera, Ruwan Jayathilaka.

**Methodology:** Yuganthi Caldera, Tharulee Ranthilake, Dilshani Senevirathna, Ruwan Jayathilaka.

**Software:** Yuganthi Caldera, Tharulee Ranthilake, Heshan Gunawardana, Ruwan Jayathilaka.

**Supervision:** Ruwan Jayathilaka, Nilmini Rathnayake, Suren Peter.

**Validation:** Yuganthi Caldera, Tharulee Ranthilake, Heshan Gunawardana, Dilshani Senevirathna, Ruwan Jayathilaka, Nilmini Rathnayake, Suren Peter.

**Visualization:** Yuganthi Caldera, Tharulee Ranthilake, Ruwan Jayathilaka.

**Writing – original draft:** Yuganthi Caldera, Tharulee Ranthilake, Heshan Gunawardana, Dilshani Senevirathna, Ruwan Jayathilaka, Nilmini Rathnayake.

**Writing – review & editing:** Ruwan Jayathilaka, Nilmini Rathnayake.

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
