## [Editor Report · Decision Letter 0]

11 Apr 2024

PONE-D-24-12923Understanding the Interplay of GDP, Renewable, and Non-Renewable Energy on Carbon Emissions: Global Wavelet Coherence AnalysisPLOS ONE

Dear Dr. Jayathilaka,

Thank you for submitting your manuscript to PLOS ONE. After careful consideration, we feel that it has merit but does not fully meet PLOS ONE’s publication criteria as it currently stands. Therefore, we invite you to submit a revised version of the manuscript that addresses the points raised during the review process.

We look forward to receiving your revised manuscript.

Kind regards,

Zhihua Zhang

Academic Editor

PLOS ONE

Journal Requirements:

2. Please note that your Data Availability Statement is currently missing the direct link to access each database. If your manuscript is accepted for publication, you will be asked to provide these details on a very short timeline. We therefore suggest that you provide this information now, though we will not hold up the peer review process if you are unable.

**Additional Editor Comments:**

The method is too simple. The authors need to demonstrate their deep understanding on wavelets, otherwise results are not reliable

it is not enough to only use one tool to do analysis. More statistical analysis should be added

---

## [Author Response · Author response to Decision Letter 0]

4 May 2024

Point by point response to editor and reviewers.

Dear Editor,

Greetings!

We extend our sincere appreciation for your invaluable comments and suggestions provided. Your insights have played a crucial role in the revision and enhancement of our work. We are genuinely grateful for the opportunity to submit the revised version of our manuscript, originally titled " Understanding the Interplay of GDP, Renewable, and Non-Renewable Energy on Carbon Emissions: Global Wavelet Coherence Analysis " to PLOS ONE, submitted on 31st March 2024.

We have diligently incorporated the suggested improvements, and the revised document includes marked modifications. For your convenience, we have prepared a table detailing the changes, with line numbers corresponding to the revised manuscript that incorporates track changes. Your continued support and guidance are highly valued, and we look forward to any further insights you or the reviewers may offer.

Please find the reference table below, aligning with the line numbers in the revised manuscript containing without track changes.

Editor’s comment 1: Please ensure that your manuscript meets PLOS ONE's style requirements, including those for file naming.

Authors’ Response to Editor's Comment 1: In response to the reviewer's comment, we have carefully reviewed PLOS ONE's style requirements and made necessary adjustments to our manuscript. Specifically, we have renamed the figures according to the journal's guidelines and ensured that each file is saved in an acceptable format as individual files. We believe these revisions align with the journal's requirements. Thank you for bringing this to our attention.

Editor’s comment 2: Please note that your Data Availability Statement is currently missing the direct link to access each database. If your manuscript is accepted for publication, you will be asked to provide these details on a very short timeline. We therefore suggest that you provide this information now, though we will not hold up the peer review process if you are unable.

Authors’ Response to Editor's Comment 2: Thank you for bringing this to our attention. We appreciate the importance of providing direct access to the data underlying our study. Below, we have included the direct links to access each database corresponding to the variables analysed in our manuscript:

1) CO2 emission in Metric tons Per Capita - https://data.worldbank.org/indicator/EN.ATM.CO2E.PC

2) Per capita GDP (Current US$) - https://data.worldbank.org/indicator/NY.GDP.PCAP.CD

3) % of total final energy consumption - https://data.worldbank.org/indicator/EG.FEC.RNEW.ZS

4) 100 - % of total final energy consumption - https://data.worldbank.org/indicator/EG.FEC.RNEW.ZS

Editor’s comment 3: The method is too simple. The authors need to demonstrate their deep understanding on wavelets, otherwise, results are not reliable,it is not enough to only use one tool to do analysis. More statistical analysis should be added.

Authors’ Response to Editor's Comment 3: We appreciate the reviewer's insightful comments regarding the methodology employed in our study. To address concerns about the robustness of our analysis, we have made significant enhancements to our methodology, particularly focusing on the utilization of cross-country Granger causality.

1) Introduction of Cross-Country Granger Causality: We have incorporated cross-country Granger causality tests to deepen our analysis and demonstrate a more comprehensive understanding of the relationships between variables. Specifically, we have examined the causality between CO2 emissions and key variables such as per capita GDP (PGDP), renewable energy (RE), and non-renewable energy (NRE). Line numbers (102 to 104) has incorporated to briefly explain the analysis.

2) Methodology Enhancement: We have provided a detailed introduction to our methodology, highlighting the incorporation of cross-country Granger causality tests. This addition can be found in the relevant section (line numbers (232 to 247).

3) Results Integration: The results of the Granger causality tests have been seamlessly integrated into our analysis, enhancing the depth and reliability of our findings. These results are presented in detail from (line numbers 632 to 685). The results obtained from the analysis is presented in S2 Appendix, S3 Appendix, S4 Appendix.

4) Incorporation into Abstract and Conclusion: To ensure transparency and coherence, we have reflected the inclusion of cross-country Granger causality tests in both the Abstract (line numbers 42 to 43) and the Conclusion (line numbers 696 to 701).

Additionally, we have introduced Violin plots to visually represent the dispersion of the data for CO2 emissions, PGDP, RE, and NRE across different country categories (line numbers 269 to 284), further enriching the analysis.

These enhancements not only address the reviewer's concerns but also contribute significantly to the rigor and depth of our study. We believe these additions strengthen the reliability and validity of our results. Thank you for the opportunity to improve our manuscript based on your valuable feedback.

---

## [Decision Letter · Decision Letter 1]

12 Jun 2024

PONE-D-24-12923R1Understanding the Interplay of GDP, Renewable, and Non-Renewable Energy on Carbon Emissions: Global Wavelet Coherence AnalysisPLOS ONE

Dear Dr. Jayathilaka,

Thank you for submitting your manuscript to PLOS ONE. After careful consideration, we feel that it has merit but does not fully meet PLOS ONE’s publication criteria as it currently stands. Therefore, we invite you to submit a revised version of the manuscript that addresses the points raised during the review process.

We look forward to receiving your revised manuscript.

Kind regards,

Zhihua Zhang

Academic Editor

PLOS ONE

Journal Requirements:

Comments from Editorial Office:

One or more of the reviewers has recommended that you cite specific previously published works. Members of the editorial team have determined that the works referenced are not directly related to the submitted manuscript. As such, please note that it is not necessary or expected to cite the works requested by the reviewer.

Reviewers' comments:

Reviewer's Responses to Questions

**Comments to the Author**

1. If the authors have adequately addressed your comments raised in a previous round of review and you feel that this manuscript is now acceptable for publication, you may indicate that here to bypass the “Comments to the Author” section, enter your conflict of interest statement in the “Confidential to Editor” section, and submit your "Accept" recommendation.

Reviewer #1: All comments have been addressed

2. Is the manuscript technically sound, and do the data support the conclusions?

Reviewer #1: Yes

3. Has the statistical analysis been performed appropriately and rigorously? 

Reviewer #1: Yes

4. Have the authors made all data underlying the findings in their manuscript fully available?

Reviewer #1: Yes

5. Is the manuscript presented in an intelligible fashion and written in standard English?

Reviewer #1: Yes

6. Review Comments to the Author

Reviewer #1: I'm pleased with the revisions made by the authors; their hard work deserves credit. I'd like to contribute my thoughts to this paper.

In Page 4, line 79 please cited the following references

Majekodunmi, T.V., Shaari, M.S., Zainal, N.F., Harun, N.H., Ridzuan, A.R., Abidin, N.Z., Abd Rahman, N. (2023). Gas consumption as a key for low carbon state and its impact on economic growth in Malaysia: ARDL approach, International Journal of Energy Economics and Policy, 13(3), 469-477. http://dx.doi.org/10.32479/ijeep.14134

Shaari, M.S., Majekodunmi, T.B., Zainal, N.F., Harun, N.H., & Ridzuan, A.R. (2023). The linkage between natural gas consumption and industrial output: New evidence based on time series analysis, Energy, 284, 129395. https://doi.org/10.1016/j.energy.2023.129395

Mohamed Yusoff, N.Y., Ridzuan, A.R., Soseco, T., Whajoedi, Narmaditya, B.S. & Ann, L.C. (2023). Comprehensive outlook on macroeconomic determinants for renewable energy in Malaysia. Sustainability, 15, 3891. https://doi.org/10.3390/su15053891

2. How does understanding causality provide novelty to your studies? Add your justification in Page 5 Line 92

3. Please add the summary of literature gap at the end of section 2: literature review (Page 9 Line 192)

4. I suggest including the explanation of your model (the foundation and the expansion) into your version. Explain the expected sign as well between iv and dv. This should be placed in the earlier part of the Methodology section, Page 10.

5. Please enhance further your policy recommendations especially what are the best policies can be suggested from the causality test.

7. PLOS authors have the option to publish the peer review history of their article (what does this mean?). If published, this will include your full peer review and any attached files.

Reviewer #1: No

---

## [Author Response · Author response to Decision Letter 1]

22 Jun 2024

Point-by-Point Response to the Academic Editor and Reviewers Comments

Dear Editor and Reviewers,

We extend our sincere appreciation for the invaluable comments provided during the initial revision round, which greatly contributed to improving the paper to its current stage. We are grateful for your dedicated time and effort, as well as the insightful feedback from the reviewers, all of which have significantly enhanced the quality of our manuscript.

We are genuinely grateful for the opportunity to submit the second revised version of our manuscript, titled “Understanding the Interplay of GDP, Renewable, and Non-Renewable Energy on Carbon Emissions: Global Wavelet Coherence and Granger Causality Analysis” to PLOS ONE, initially submitted on 31st March 2024. We have diligently incorporated the suggested improvements, and the revised document includes clearly marked modifications. 

As suggested, we have prepared a table detailing the changes, with line numbers corresponding to the revised manuscript that incorporates without track changes as follows. Your continued support and guidance are highly valued, and we look forward to any further insights you or the reviewers may offer.

Please find the reference table below, aligning with the line numbers in the revised manuscript containing without track changes. 

Journal Requirements

Requirement #1: Please review your reference list to ensure that it is complete and correct. If you have cited papers that have been retracted, please include the rationale for doing so in the manuscript text, or remove these references and replace them with relevant current references. Any changes to the reference list should be mentioned in the rebuttal letter that accompanies your revised manuscript. If you need to cite a retracted article, indicate the article’s retracted status in the References list and also include a citation and full reference for the retraction notice.

Response by Authors to Requirement #1:

We sincerely appreciate your careful review and valuable feedback on the reference list. 

We have thoroughly reviewed our citations and made necessary revisions. Upon rechecking our reference list, we identified that reference ("Zhang et al., 2023") had been retracted from the journal due to a serious breach of journal authorship policies and of publication ethics. We regret any oversight in not recognising this retraction earlier. We have promptly removed the retracted reference and replaced it with a relevant current reference:

70. Amin, N. and Song, H. (2022) ‘The role of renewable, non-renewable energy consumption, trade, economic growth, and urbanization in achieving carbon neutrality: A Comparative Study for South and East Asian countries’, Environmental Science and Pollution Research, 30(5), pp. 12798–12812. doi:10.1007/s11356-022-22973-2.

Additionally, we have ensured that all changes to the reference list are clearly documented in the revised manuscript with the track changes.

Comments from Editorial Office

Editor’s comment 1: One or more of the reviewers has recommended that you cite specific previously published works. Members of the editorial team have determined that the works referenced are not directly related to the submitted manuscript. As such, please note that it is not necessary or expected to cite the works requested by the reviewer.

Response by Authors to Editor’s Comment #1: We appreciate the editorial team's clarification and thank you for your feedback regarding the reviewer's recommendation to cite specific previously published works in our manuscript.

We have thoroughly analysed the suggested published works to cite with the scope of our study and decided to cite one most relevant published work [Mohamed Yusoff et al. (2023)] for the current study (Refer from line number 80 to 85) by refraining from incorporating the rest of the two published works [Majekodunmi et al. (2023) and Shaari et al. (2023)] which are not directly align with the comprehensive scope of our study.

Review Comments to the Author

Reviewer #1 Comment 1: I'm pleased with the revisions made by the authors; their hard work deserves credit. I'd like to contribute my thoughts to this paper.

In Page 4, line 79 please cited the following references.

Majekodunmi, T.V., Shaari, M.S., Zainal, N.F., Harun, N.H., Ridzuan, A.R., Abidin, N.Z., Abd Rahman, N. (2023). Gas consumption as a key for low carbon state and its impact on economic growth in Malaysia: ARDL approach, International Journal of Energy Economics and Policy, 13(3), 469-477. http://dx.doi.org/10.32479/ijeep.14134

Shaari, M.S., Majekodunmi, T.B., Zainal, N.F., Harun, N.H., & Ridzuan, A.R. (2023). The linkage between natural gas consumption and industrial output: New evidence based on time series analysis, Energy, 284, 129395. https://doi.org/10.1016/j.energy.2023.129395

Mohamed Yusoff, N.Y., Ridzuan, A.R., Soseco, T., Whajoedi, Narmaditya, B.S. & Ann, L.C. (2023). Comprehensive outlook on macroeconomic determinants for renewable energy in Malaysia. Sustainability, 15, 3891. https://doi.org/10.3390/su15053891

Response by Authors to Reviewer Comment 1: Thank you for your kind words and encouragement regarding the revisions made to our manuscript. We sincerely appreciate your feedback and insightful comments, which have played a pivotal role in improving our manuscript to its current stage.

Thank you for your suggestion regarding the citation of references.

After careful consideration and in alignment with the editorial team’s guidance, we have reviewed the suggested references. We have decided to cite the most relevant work, Mohamed Yusoff et al. (2023), which discusses macroeconomic determinants for renewable energy, as it aligns closely with the comprehensive scope of our study (Refer from Line number 80 to 85).

The paragraph added is, “In Malaysia, several studies have implied the significant role of natural gas consumption in driving economic growth while promoting environmental sustainability. Natural gas, comparing to oil and coal, emits lower levels of CO2, positioning it as a crucial energy source for mitigating environmental degradation and supporting transitions towards cleaner energy”. Regrettably, we have refrained from incorporating the remaining references, Majekodunmi et al. (2023) and Shaari et al. (2023), as they do not directly relate to the specific variables and global analysis discussed in our manuscript.

We appreciate your understanding and constructive feedback throughout this process. Your insights have been instrumental in shaping our manuscript.

Reviewer #1 Comment 2: How does understanding causality provide novelty to your studies? Add your justification in Page 5 Line 92

Response by Authors to Reviewer Comment 2: Response by Authors to Reviewer Comment 2

Thank you for bringing this point to our attention. We recognise the importance of adding the understanding of causality to provide the novelty to our study. 

We have incorporated the following paragraph in page 5 from line number 99 - 111 to address this concern. 

“Understanding the causality enhances the novelty of our study by revealing the intricate relationship between economic growth, renewable and non - renewable energy consumption, and CO2 emissions trough sophisticated techniques like wavelet coherence and Granger causality analysis. This approach transcends simple correlation, offering a dynamic perspective on how these variables interact over time. By differentiating between unidirectional and bidirectional causality patterns and examining dynamics across global income levels, our study fills a substantial gap in the literature. It equips policy makers with actionable insights to design targeted interventions for sustainable development and climate mitigation efforts. This comprehensive understanding of causality dynamics significantly enriches the existing body of knowledge and enhances the effectiveness of policy making in addressing pressing environmental challenges.”

We believe this clarification explain the novel contributions of our research effectively.

Reviewer #1 Comment 3: Please add the summary of literature gap at the end of section 2: literature review (Page 9 Line 192)

Response by Authors to Reviewer Comment 3: Thank you for your suggestion regarding the addition of a summary of the literature gap.

We appreciate the importance of clearly delineating the gaps in existing literature. 

As suggested, we have added a summary of the literature gap at the end of section 2 (Refer the line number 216 to 230 in page 9 and page 10).

The included paragraph as follows,

“In culmination of the literature review, critical lacunae emerge in the existing body of research, notably pertaining to the dearth of studies that systematically categorise economic pathways across income levels using wavelet methodologies. This gap underscores the necessity for comprehensive analyses that not only employ advanced analytical techniques but also encompass diverse country contexts to elucidate the exact dynamics between economic growth, energy consumption and CO2 emissions. Furthermore, despite the considerable scholarly attention devoted to the Environmental Kuznets Curve hypothesis and the relationship between economic growth and CO2 emissions, there persists a notable absence of comprehensive assessments that encompass temporal dynamics and diverse country contexts. Additionally, the literature review reveals a conspicuous gap in the examination of the role of renewable energy utilization in mitigating CO2 emissions, particularly in regions abundant in renewable resources. Addressing these gaps is imperative to inform evidence-based policy interventions aimed at fostering sustainable development and mitigating adverse effects of climate change.”

Reviewer #1 Comment 4: I suggest including the explanation of your model (the foundation and the expansion) into your version. Explain the expected sign as well between iv and dv. This should be placed in the earlier part of the Methodology section, Page 10.

Response by Authors to Reviewer Comment 4: Thank you for your insightful suggestion. When analysing the causality, we have analysed the expected sign for the variables based on the previous studies. 

We have revised the manuscript to incorporate an explanation of our model, including the expected signs between independent variables (IV) and dependent variables (DV), with a supporting table (Table 2) to give more understating. 

Kindly refer to the line number 254 to 260 in page 11 & 12.

Reviewer comment 05: Please enhance further your policy recommendations especially what are the best policies can be suggested from the causality test.

Response by Authors to Reviewer Comment 05: Thank you for your valuable suggestion. 

In response, we have revised the manuscript by expanding the policy recommendations section to include actionable measures derived from the causality analyses. These include promoting renewable energy adoption, investing in sustainable technologies, strengthening regulatory frameworks, implementing urban transportation policies to reduce emissions and arranging international collaborations to address global environmental challenges.

 Kindly refer to the line number 775 to 801 in page 44 & 45.

We hope these enhancements to the policy recommendations section based on the causality test of the current study will provide valuable insights to guide policymakers and researchers toward sustainable decision making.

---

## [Decision Letter · Decision Letter 2]

31 Jul 2024

Understanding the Interplay of GDP, Renewable, and Non-Renewable Energy on Carbon Emissions: Global Wavelet Coherence and Granger Causality Analysis

PONE-D-24-12923R2

Dear Dr. Jayathilaka,

We’re pleased to inform you that your manuscript has been judged scientifically suitable for publication and will be formally accepted for publication once it meets all outstanding technical requirements.

Kind regards,

Zhihua Zhang

Academic Editor

PLOS ONE
---

## [Editor Report · Acceptance letter]

5 Aug 2024

PONE-D-24-12923R2 

PLOS ONE

Dear Dr. Jayathilaka, 

I'm pleased to inform you that your manuscript has been deemed suitable for publication in PLOS ONE. Congratulations! Your manuscript is now being handed over to our production team.

Kind regards, 

on behalf of

Dr. Zhihua Zhang 

Academic Editor

PLOS ONE